# Technical note: *in situ* measurement of flux and isotopic composition of $CO_2$ released during oxidative weathering of sedimentary rocks

Guillaume Soulet[1], Robert G. Hilton[1], Mark H. Garnett[2], Mathieu Dellinger[1], Thomas Croissant[1], Mateja Ogrič[1] and Sébastien Klotz[3]

[1]Department of Geography, Durham University, South Road, Durham DH1 3LE, United Kingdom
[2]NERC Radiocarbon Facility, Rankine Avenue, East Kilbride, Glasgow G75 0QF, UK
[3]IRSTEA Grenoble, 2 rue de la papeterie, BP 76, 38402 Saint-Martin-d'Hères, Cedex, France

*Correspondence to*: Guillaume Soulet (guillaume.s.soulet@durham.ac.uk)

**Abstract.** Oxidative weathering of sedimentary rocks can release carbon dioxide ($CO_2$) to the atmosphere and is an important natural $CO_2$ emission. Two mechanisms operate – the oxidation of sedimentary organic matter and the dissolution of carbonate minerals by sulphuric acid. It has proved difficult to directly measure the rates at which $CO_2$ is emitted in response to these weathering processes in the field, with previous work generally using methods which track the dissolved products of these reactions in rivers. Here we design a chamber method to measure $CO_2$ production during the oxidative weathering of shale bedrock, which can be applied in erosive environments where rocks are exposed frequently to the atmosphere. The chamber is drilled directly into the rock face and has a high surface area to volume ratio which benefits measurement of $CO_2$ fluxes. It is a relatively low cost method and provides a long-lived chamber (several months or more). To partition the measured $CO_2$ fluxes and the source of $CO_2$, we use zeolite molecular sieves to trap $CO_2$ 'actively' (over several hours) or 'passively' (over a period of months). The approaches produce comparable results, with the trapped $CO_2$ having a radiocarbon activity (Fraction modern, Fm) ranging from Fm = 0.05 to Fm = 0.06 and demonstrating relatively little contamination from local atmospheric $CO_2$ (Fm = 1.01). We use stable carbon of the trapped $CO_2$ to partition between an organic and inorganic carbon source. The measured fluxes of rock-derived organic matter oxidation ($171\pm5$ mgC.m$^{-2}$.day$^{-1}$) and carbonate dissolution by sulphuric acid ($534\pm16$ mgC.m$^{-2}$.day$^{-1}$) from a single chamber were high when compared to the annual flux estimates derived from using dissolved river chemistry in rivers around the world. The high oxidative weathering fluxes are consistent with the high erosion rate of the study region. We propose our in situ method has the potential to be more widely deployed to directly measure $CO_2$ fluxes during the oxidative weathering of sedimentary rocks, allowing for the spatial and temporal variability in these fluxes to be determined.

## 1 Introduction

The stock of carbon contained within sedimentary rocks is vast, with $\sim1.25\times10^7$ PgC contained within organic matter and $\sim6.53\times10^7$ PgC as carbonate minerals (Sundquist and Visser, 2005). If these rocks are exposed to Earth's oxygenated surface, for instance during rock uplift, erosion and exhumation, oxidative weathering can result in a release of carbon dioxide ($CO_2$)

from the lithosphere to the atmosphere (Petsch et al., 2000). There are two main processes to consider: i) the oxidation of rock-derived organic carbon (Berner and Canfield, 1989; Petsch, 2014), which can be expressed by the (geo)respiration of organic matter:

$$CH_2O + O_2 \rightarrow CO_2 + H_2O \tag{1}$$

and ii) the oxidation of sulphide minerals (e.g., pyrite) which produces sulphuric acid, which can chemically weather carbonate minerals and release $CO_2$ (Calmels et al., 2007; Li et al., 2008; Torres et al., 2014) by the reaction:

$$CaCO_3 + H_2SO_4 \rightarrow CO_2 + H_2O + Ca^{2+} + SO_4^{2-} \tag{2}$$

or

$$2CaCO_3 + H_2SO_4 \rightarrow 2Ca^{2+} + 2HCO_3^- + SO_4^{2-} \rightarrow CaCO_3 + CO_2 + H_2O + Ca^{2+} + SO_4^{2-} \tag{3}$$

In the case of Eq. (1) and Eq. (2), $CO_2$ is released to the atmosphere at the site of chemical weathering. In the case of Eq. (3), $CO_2$ is released to the atmosphere over a timescale equivalent to that of the precipitation of carbonate in the ocean (~$10^4$ to $10^6$ years; Berner and Berner, 2012).

The fluxes of carbon transferred to the atmosphere in response to both oxidative weathering processes are thought to be as large as that released by volcanic degassing, but the absolute fluxes remain uncertain (Li et al., 2008; Petsch, 2014). As such, both processes act to govern the levels of atmospheric $CO_2$ and $O_2$, and hence Earth's climate over geological timescales (Berner and Canfield, 1989; Torres et al., 2014). The oxidation of rock-derived organic carbon may also contribute to modern biological cycles, especially rock substrate that is rich in organic carbon (Bardgett et al., 2007; Copard et al., 2007; Keller and Bacon, 1998; Petsch et al., 2001). Various approaches have been adopted to better quantify these major geological $CO_2$ sources. These include the use of geochemical proxies in rivers, which indirectly track the $CO_2$ emissions from the oxidative weathering of sedimentary rocks at the catchment-scale. For instance, the trace element rhenium has been used to compare relative rates of rock-derived organic carbon oxidation (Jaffe et al., 2002) and estimate the corresponding fluxes of $CO_2$ across river catchments (Dalai et al., 2002; Hilton et al., 2014; Horan et al., 2017). Another approach has been to measure the loss of radiocarbon-depleted organic matter in river sediments during their transfer across the floodplains of large river basins (Bouchez et al., 2010; Galy et al., 2008). In the case of sulphuric acid-weathering of carbonate minerals, the dissolved sulphate flux can be informative if the source of $SO_4^{2-}$ has been assessed using sulphur and oxygen isotopes (Calmels et al., 2007; Hindshaw et al., 2016; Spence and Telmer, 2005) and/or using the dissolved inorganic carbon flux and its stable carbon isotope composition ($\delta^{13}C$) (Galy and France-Lanord, 1999; Li et al., 2008; Spence and Telmer, 2005).

It should be possible to directly measure the flux of $CO_2$ emanating from sedimentary rocks in response to oxidative weathering. Keller and Bacon (1998) explored such an approach in a 7 m deep soil on till, suggesting geo-respiration of

Cretaceous age organic matter was an important source of $CO_2$ at depth. However, this research has not to our knowledge been replicated, nor applied in erosive landscapes where sedimentary rocks are frequently exposed to weathering by erosion processes (Blair et al., 2003; Hilton et al., 2011). In these locations, oxidative weathering fluxes are thought to be very high (Calmels et al., 2007; Hilton et al., 2014; Petsch et al., 2000). One of the challenges of tracking $CO_2$ directly is that flux

measurements must be combined with the isotopic composition ($^{12}C$, $^{13}C$ and $^{14}C$) of the $CO_2$ (Keller and Bacon, 1998). Only with that information can the measured $CO_2$ flux be partitioned into the component derived from the oxidation of rock-derived carbon and that derived from the dissolution of carbonate (in addition to inputs from the modern plant and soil biosphere, and atmospheric inputs).

The objective of this paper is to provide a detailed proof of concept study of methods we have designed which can: (1) make

direct measurements of the flux of $CO_2$ released during the oxidative weathering of sedimentary rocks; and (2) trap the $CO_2$ produced during weathering in order to measure its isotope composition, and partition the source of the $CO_2$ flux between rock-derived organic carbon and carbonate. Here we outline one approach to address these research gaps which adapts a chamber-based method to measure $CO_2$ fluxes. We provide the first examples of its application to trap $CO_2$ and use the isotope composition to directly quantify the fluxes of $CO_2$ from oxidative weathering.

## 2 Methods

### 2.1 Study area

The study location is within the Laval catchment, part of the IRSTEA Draix Bléone Experimental Observatory and the Réseau des Bassins Versants network, located near the town of Draix, Alpes de Haute Provence, France. The small catchment (0.86 $km^2$) is heavily instrumented, with continuous monitoring of rainfall, river water discharge, river solid load transport, total

dissolved fluxes and physical erosion rates over the last four decades (Cras et al., 2007; Mathys et al., 2003). These measurements provide hydrodynamic and geomorphic context for any studies of oxidative weathering. The lithology of the catchment is composed of black Jurassic marine marls (from the Bajocian to the Callovo-Oxfordian stages), which contain inorganic carbon and organic matter (Graz et al., 2012). Sulphide minerals are widespread as disseminated pyrite and veins which outcrop in the catchment (Cras et al., 2007). The rock-strength, climate and geomorphic setting combine to produce a

badland-type morphology with very steep and dissected slopes.

Erosion rates are very high, with sediment export fluxes of 11,200 tons.$km^{-2}$.$yr^{-1}$ on average during the period 1985 to 2005 AD (with a minimum of 4,400 tons.$km^{-2}$.$yr^{-1}$ in 1989 and a maximum of 21,100 tons.$km^{-2}$.$yr^{-1}$ in 1992) (Graz et al., 2012; Mathys et al., 2003). Assuming a regolith bulk density of ~1 to 1.5 tons.$m^{-3}$ (Mathys and Klotz, 2008; Oostwoud Wijdenes and Ergenzinger, 1998), this equates to a physical erosion rate of ~7 to 10 mm.$yr^{-1}$ on average, but that can reach maximum

values of 20 to 14 mm.$yr^{-1}$. These features limit the development of soils and bare rock outcrops represent 68% of the catchment surface area (i.e., 0.58 $km^2$) (Cras et al., 2007; Mathys et al., 2003). As a result, it is easy to find regolith and rock surfaces that are devoid of soils and roots, and where sedimentary rocks are directly exposed to the oxygen-rich atmosphere. These are key

parts of the landscape contributing to weathering, solute production (Cras et al., 2007) and sediment production (Graz et al., 2012; Mathys et al., 2003; Oostwoud Wijdenes and Ergenzinger, 1998). Bare rock outcrops are characterized by the development of weathered marls and regolith. Regolith typically extends at least to ~20 cm with the following characteristics: i) the upper ~3 cm is loose detrital cover composed of mm-to-cm fragments of marls, ii) from ~3 to ~10cm is the loosened upper regolith which is somewhat fragmented, iii) from ~10 to 20 cm is the compact lower regolith which retains the marl structure but not its cohesion, and iv) at depths more than ~20cm is the marl bedrock (unweathered marl) (Maquaire et al., 2002; Mathys and Klotz, 2008; Oostwoud Wijdenes and Ergenzinger, 1998). Lateral variation in the regolith thickness exists with larger thicknesses on crests, intermediate in gullies and minimal in talwegs (Maquaire et al., 2002). Marl bedrock porosity ranges between 0.17 and 0.23 (Traveletti et al., 2012).

## 2.2 Natural oxidation and $CO_2$ accumulation chambers

In order to measure the flux of $CO_2$ produced by oxidative weathering of sedimentary rocks, and accumulate enough $CO_2$ to perform stable carbon isotope and radiocarbon measurements, we use accumulation chambers (e.g., Billett et al., 2006; Hardie et al., 2005). These have been extensively used to measure soil respiration (e.g., Hahn et al., 2006; Hardie et al., 2005), $CO_2$ evasion by streams and rivers (e.g., Billett et al., 2006; Borges et al., 2004), but have not yet been used to examine rock-atmosphere interactions. Because most fine-grained sedimentary rocks have a degree of competency, accumulation chambers can be made by directly drilling into the rock. Here we use a rock-drill to make 40 cm-long chambers with an inner diameter of 29 mm. Our aim was to minimise the volume of the chamber while maximizing the surface of exchange with the surrounding rock.

The rock powder left inside the chamber after its drilling was blown away using a compressed-air gun in order to minimize the presence of potentially reactive dust. Then, after measuring the dimensions of the chamber, its entrance is fitted with a small PVC tube (~3 cm-diameter, ~3 cm-long), which allows a tight seal with an inserted rubber stopper containing two holes. Two Pyrex® tubes (ID=3.4 mm and OD=5 mm; one of 12 cm-long and one of 7 cm-long) are inserted through the rubber stopper. The differential length is to improve airflow in the chamber while in operation. The sections of the Pyrex® tubes sticking out of the chamber are fitted with Tygon® tubing (E-3603; Saint Gobin, France). To isolate the accumulation chamber from the atmosphere as best as possible, the Tygon® tubing is sealed with WeLock® clips (Scandinavia Direct Ltd, UK) and silicone sealant (Unibond® Outdoor) is placed around the entrance of the chamber (the 3cm-diameter PVC tubing and the rubber stopper) (Figure 1). The newly installed chamber is left for ~2 days to allow the sealant to fully dry. Here we acknowledge that a perfect seal is impossible, due to the natural rock fracturing around the chamber. Table 1 summarises the dimensions of an example chamber drilled and sealed in the field on 13[th] December 2016.

Drilling introduces an oxygen-rich atmosphere in the chamber and surrounding marl regolith and bedrock (similar to outcropping marls exposed to the atmosphere). If gaseous $O_2$ is consumed (e.g. by Eq. (1)), this would create a gradient in the partial pressure of $O_2$ ($pO_2$) whereby the atmosphere surrounding the rock and chamber is of higher $pO_2$. Given the natural porosity and permeability of the shale bedrock, any diffusion of $O_2$ is likely to be into the chamber. This should act continuously

to fuel the chamber with oxygen. In contrast, if $CO_2$ is produced inside the chamber (by Eq. (1) and Eq. (2)) then the partial pressure of $CO_2$ ($pCO_2$) will exceed that of the atmosphere. The result is that for chambers where oxidative weathering is occurring, the ingress of 'contamination' by atmospheric $CO_2$ should be minimal, and there should be a supply of $O_2$ for reactions. These inferences can be tested using a $pO_2$ probe and by trapping $CO_2$ and measuring its isotope composition.

In this example we aimed to measure oxidation of sedimentary rocks, and intended to minimise the role of $CO_2$ produced by root respiration. Therefore, the chambers were drilled on cleared rock surface, devoid of visible roots. The rock powder produced when drilling the chambers was collected, freeze-dried and grinded in the laboratory to fine powder for measurement of its organic-inorganic carbon content and its isotopic composition.

## 2.3 $CO_2$ flux measurements

A closed-loop $CO_2$ sampling system similar to the molecular sieve sampling system ($MS^3$) described in Hardie et al. (2005) was used for $CO_2$ flux measurements and $CO_2$ sampling (Figure 1). The system incorporated the following components: an air filter, a water trap (cartridge filled with magnesium perchlorate), a portable infrared gas analyser (IRGA) equipped with an internal air pump, calibrated to a $pCO_2$ range of 0 to 5000 ppmv and installed with an $pO_2$ probe (EGM-5, PP Systems, US), a $CO_2$ scrub (cartridge filled with soda lime), a bypass, and a set of WeLock® clips that allows the air flow to be diverted through

the bypass or the $CO_2$ scrubber cartridge. Optionally a zeolite molecular sieve sampling cartridge can be inserted in the line (see next section).

Before each $CO_2$ flux measurement, the Tygon® tubes exiting the chamber were fitted with autoshutoff Quick Couplings™ (Colder Products Company, USA), and the $CO_2$ contained within the sampling system removed using the $CO_2$ scrubber cartridge. When no $CO_2$ is left in the sampling system (as indicated by the IRGA), the air flow is diverted through the bypass,

and the system connected to the chamber (Figure 2). The use of the auto-shutoff couplings prevents atmospheric contamination at the moment of connection to the chamber. Then, $pCO_2$ in the chamber is lowered to near atmospheric $pCO_2$ by guiding the air flow through the $CO_2$ scrubber cartridge. We let the $CO_2$ accumulate in the chamber for several minutes (typically 10 minutes) by guiding the air flow through the bypass (Figure 2). This operation can be repeated several times to provide multiple measurements of $CO_2$ flux over a period of hours (Figure 3). The $CO_2$ evolution in the chamber typically shows a curvature,

the curve flattening with time and higher concentration (Figure 3). In order to calculate the $CO_2$ flux, we first convert the $pCO_2$ measurements into the mass of carbon contained in the chamber:

$$m = \frac{pCO_2}{10^6} \cdot V \cdot A \tag{4}$$

where m is the mass of carbon in the chamber (in mgC), $pCO_2$ the concentration of $CO_2$ in the chamber in ppm ($cm^3.m^{-3}$), V is the total volume ($cm^3$) i.e., the sum of the volume of the chamber ($V_{Ch}$) and the volume of the $CO_2$ sampling system ($V_L$)

when air flows through the bypass. Factor A converts centimetres cubed of $CO_2$ into milligrams of carbon, depending on temperature and pressure following the ideal gas law:

$$A = \frac{P \cdot M_C}{R \cdot T} \cdot 10^{-3} \tag{5}$$

where P is the pressure (Pa) as measured by the IRGA, $M_C$ is the molar mass of carbon (g.mol$^{-1}$), R is the gas constant (m$^3$.Pa.K$^{-1}$.mol$^{-1}$) and T the temperature (K) in the chamber. Then the rate (q in mgC.min$^{-1}$) at which carbon accumulates in the chamber is calculated using an exponential model (described below; Pirk et al., 2016) and converted into a flux of carbon (Q in mgC.m$^{-2}$.day$^{-1}$) emitted to the atmosphere under the form of $CO_2$ using:

$$Q = 1440 \, q/S \tag{6}$$

where 1440 converts mgC.min$^{-1}$ into mgC.day$^{-1}$, and S (m$^2$) is the inner surface area of the chamber exchanging with the surrounding rock. To calculate the rate of accumulation of carbon (q) in the chamber we use the exponential model described by Pirk et al. (2016):

$$\frac{dm(t)}{dt} = q - \lambda(m(t) - m_0) \tag{7}$$

where $\frac{dm(t)}{dt}$ is the carbon mass change in the chamber with time. Parameter $m_0$ is the mass of carbon in the chamber at the beginning of the $CO_2$ accumulation and that should be close to the mass of carbon in the chamber at atmospheric $pCO_2$. The constant $\lambda$ (in units of time$^{-1}$, here in min$^{-1}$) describes the sum of all processes which are proportional to the carbon mass difference $\Delta m(t) = m(t) - m_0$ and is responsible for the curvature of the carbon mass accumulation evolution (Figure 3). The model does not a priori assume any process to be responsible for the curvature (Pirk et al., 2016). In the case of the measurement of $CO_2$ flux in soils, the curvature ($\lambda$) relates to leakages, diffusivity from soil $CO_2$ into the chamber headspace and photosynthesis (Kutzbach et al., 2007). In the case of our chambers drilled in rock, since it is assumed that there is no possibility of photosynthesis, $\lambda$ likely relates to the diffusivity of carbon from the rock to the chamber headspace and to the chamber leakiness. Equation Eq. (7) is solved by fitting the following function to the data (Figure 3B):

$$m(t) = \frac{q}{\lambda}\left(1 - \exp(-\lambda t)\right) + m_0 \tag{8}$$

Several parameters lead to uncertainties in the flux calculations. They are all related to the conversion of $pCO_2$ to mass of carbon (Table 1): i) the volume of the chamber ($V_{Ch}$); ii) the surface area of exchange with the surrounding rock (S); iii) the volume of the closed-loop system when air flows through the bypass ($V_L$ was determined to be $127.8 \pm 0.5$ cm$^3$ through an experiment of successive $CO_2$ dilution in a known volume); and iv) the temperature in the chamber was assumed to range from 0 to 20ºC over the course of the experiment. We estimated the relative uncertainty on the measured flux using a Monte-Carlo simulation of error propagation using the ranges listed above and in Table 1. The resulting relative uncertainty on the measured flux was estimated to be within $\pm$ 2.5 %. An additional relative uncertainty linked to the rate of $CO_2$ accumulation in the chamber (parameter q obtained through fitting the exponential model to the data) ranges between 0.5 to 1.0 %. Altogether, the

final relative uncertainty determined with our Monte-Carlo simulation of error propagation was found to be within ± 3 %. In the case that the relative standard deviation on multiple flux measurements is higher than 3 %, we adopt the standard deviation as the uncertainty.

## 2.4 $CO_2$ sampling and isotopic analysis

$CO_2$ evading the rock accumulates in the chamber and can be sampled using a zeolite molecular sieve trap (Garnett et al., 2009; Garnett and Hardie, 2009; Hardie et al., 2005). Zeolites have a high affinity for polar molecules such as $H_2O$ and $CO_2$, and are widely used to separate $CO_2$ from air at ambient temperature and pressure. The gas trapped by the zeolite sieve can be extracted in the laboratory at high temperature for $CO_2$ purification and isotope analysis (Garnett and Murray, 2013; Hardie et al., 2005). The type of zeolite (13X) and the design of the cartridge containing the zeolite, is described by Hardie et al. (2005)
and Garnett et al. (2009). In our study the $CO_2$ was sampled 'actively' – i.e., using the $CO_2$ sampling system coupled to the pump incorporated in the IRGA to force the air through the zeolite molecular sieve cartridge (Figure 1) following Hardie et al. (2005). Two approaches can be used. The first involves connection of the line to the chamber for the duration of trapping, which was used on 27/03/2017. Each removal of $CO_2$ onto the trap can be controlled to return the chamber to ambient atmospheric $pCO_2$, allowing for a subsequent measurement of $CO_2$ flux (Figure 3). The second approach allows $pCO_2$ to
accumulate in the chamber, before attaching the scrubbed line and removing the $CO_2$, which we tested on 30/03/2017. The benefit of the latter method is that it allows the gas line and IRGA to be used for other tasks while in the field, but may be more susceptible to atmospheric inputs during the connection of lines.

  The $CO_2$ was also sampled 'passively', when the zeolite molecular sieve is connected to the chamber for several months following the procedure described in Garnett et al. (2009) (Figure 4). This approach has the benefit of providing an integrative
view of $CO_2$ production over longer periods of time. A passive trap was installed on the 15th December 2017 (2 days after the chamber was constructed) and removed on the 26th March 2017 (101 days after its installation) for chamber H6. Based on previous work (Garnett et al., 2009; Garnett and Hardie, 2009; Garnett and Hartley, 2010), it is expected that the passive trap method can lead to a fractionation of the stable carbon isotope composition ($\delta^{13}C$) of $4.2 \pm 0.3$ ‰ associated with the diffusion of $CO_2$ from the chamber to the zeolite trap. In addition, a sample of local atmospheric $CO_2$ was also collected by actively
circulating the atmosphere sampled at ~ 3m elevation above the valley floor through a zeolite molecular sieve.

  After sample collection the zeolite molecular sieves were sealed with WeLock® clips on either end before being disconnected from the sampling system (active or passive) and returned to the NERC Radiocarbon Facility (East Kilbride, UK) for $CO_2$ recovery. The $CO_2$ collected was desorbed from the zeolite by heating (425ºC) and cryogenically purified (Garnett and Murray, 2013). One aliquot of the recovered $CO_2$ was used for stable carbon isotope composition ($\delta^{13}C$) measurement using Isotope
Ratio Mass Spectrometry (IRMS; Thermo Fisher Delta V; results expressed relative to the Vienna Pee Dee Belemnite (VPDB) standard). A further aliquot was converted to graphite and analysed for $^{14}C/^{12}C$ ratio using accelerator mass spectrometry at the Scottish Universities Environmental Research Centre (SUERC). Radiocarbon measurements were, following convention, corrected for isotopic fractionation using the measured sample IRMS $\delta^{13}C$ values, and reported in the form of the fraction

modern (Fm) [$A_{SN}/A_{ON}$ in Stuiver and Polach (1977); corresponding to $^{14}a_N$ in Mook and van der Plicht (1999), or $F^{14}C$ in Reimer et al. (2004)] (Table 2).

## 2.5 Partitioning the sources of $CO_2$

As the chamber was drilled away from the obvious influence of root respiration, the $CO_2$ emitted from the rock should originate from: i) the oxidation of the organic carbon contained within the rock mass following Eq. (1); and/or from ii) the dissolution of the carbonate minerals by sulphuric acid following Eq. (2). Some of the $CO_2$ collected in active or passive zeolite molecular sieves might also originate from atmospheric $CO_2$, i.e., the ambient air (see discussion below). To correct for possible atmospheric contamination, and partition the sources of $CO_2$, we solve the following isotope-mass balance system:

$$\begin{cases} f_{Atm} + f_{Rock\,OC} + f_{Carb} = 1 \\ f_{Atm} \cdot \delta^{13}C_{Atm} + f_{Rock\,OC} \cdot \delta^{13}C_{Rock\,OC} + f_{Carb} \cdot \delta^{13}C_{Carb} = \delta^{13}C_{Chamber} \\ f_{Atm} \cdot Fm_{Atm} + f_{Rock\,OC} \cdot Fm_{Rock\,OC} + f_{Carb} \cdot Fm_{Carb} = Fm_{Chamber} \end{cases} \qquad (9)$$

where, $f_{Atm}$ is the mass fraction of $CO_2$ originating from the atmosphere, $f_{Rock\,OC}$ is that originating from the oxidation of the rock-derived organic carbon and, $f_{Carb}$ is that originating from carbonate dissolution by sulphuric acid. Subscript "Chamber" stands for the $CO_2$ sampled from the chambers (i.e., trapped in the zeolite molecular sieves). $\delta^{13}C$ and Fm stand for the stable carbon isotope and radiocarbon compositions of the three possible sources of $CO_2$ listed above and of the $CO_2$ sampled in the chamber.

Table 3 shows the $\delta^{13}C$ and Fm values of the three possible sources of $CO_2$ involved in the isotope-mass balance. These values were measured in the laboratory. The $\delta^{13}C_{Atm}$ and $Fm_{Atm}$ values were measured from the atmospheric $CO_2$ sample actively trapped in a zeolite molecular sieve (see Sect. 2.4.). The stable carbon isotope composition of the rock-derived organic carbon ($\delta^{13}C_{Rock\,OC}$) was obtained by IRMS after inorganic carbon removal from the rock powdered samples by HCl fumigation, followed by closed-tube combustion to produce $CO_2$. The stable carbon isotope composition of the carbonates ($\delta^{13}C_{Carb}$) was obtained after dissolution of the carbonates of the rock powdered samples by phosphoric acid in vacuumed vessels following standard procedures at NERC Radiocarbon Facility. Since the rock-derived organic carbon and carbonates were formed millions of years ago they do not contain radiocarbon any longer, and their fraction modern ($Fm_{Rock\,OC}$ and $Fm_{Carb}$) levels should be close to the AMS background as confirmed by our measurements (Table 3). Consequently, when solving the isotope-mass balance, $Fm_{Rock\,OC}$ and $Fm_{Carb}$ were set to 0.

## 3 Results and Discussion

Here we present the results (Table 2, 3, 4 and Figures 2 and 3) obtained from a natural weathering chamber (H6) drilled in a rock face at 2 meters elevation above the Laval stream (Figure 1) in December 2016. Our aim is to assess the feasibility of the method, in terms of: i) measuring the fluxes of $CO_2$; ii) collecting sufficient mass of $CO_2$ for isotope analysis (to partition

between organic and inorganic derived $CO_2$); and iii) checking the role of atmospheric $CO_2$ contamination for both the active and passive $CO_2$ sampling methods. We discuss the results from chamber H6 in the context of using this method more widely to better quantify rates of oxidative weathering and associated $CO_2$ release.

## 3.1 Flux measurements

Three months after the installation of the chamber H6, $CO_2$ fluxes were measured alongside a series of zeolite-trapping events on 27/03/2017 (Figure 3). If the chamber was perfectly isolated from the atmosphere, then we might expect the rate of carbon accumulation ($\frac{dm(t)}{dt}$) to be constant, while it decreases with time. As expected, this indicates that the chamber is not perfectly sealed. This has some important implications. First, the leak rate depends on the $pCO_2$ gradient between the chamber and the atmosphere. Since this gradient is positive in the chamber (Figure 2) ($pCO_{2_{chamber}} > pCO_{2_{atmosphere}}$), then $CO_2$ likely

diffuses from the chamber to the atmosphere. This has the advantage that it naturally minimizes any atmospheric $CO_2$ contamination. Conversely, since the $CO_2$ production is linked to the consumption of $O_2$, then the $O_2$ gradient is expected to be negative ($pO_{2_{chamber}} < pO_{2_{atmosphere}}$), and thus atmospheric $O_2$ naturally diffuses inside the chamber. This means that the chamber can be closed for months and still contain gaseous $O_2$. Our measurements of $O_2$ using the EGM-5 $O_2$ probe suggest that the chamber had a similar $pO_2$ as that contained in the ambient atmosphere of the catchment (the chamber value was 96

to 99% of the atmosphere $pO_2$).

    The fluxes of $CO_2$ measured in this chamber on 27/03/2017 decreased from $1384 \pm 42$ mgC.m$^{-2}$.day$^{-1}$ to $684 \pm 21$ mgC.m$^{-2}$.day$^{-1}$ with the number of times we extracted the $CO_2$ from the chamber (Figure 3). The flux becomes approximately constant after three $CO_2$ extractions during zeolite trapping, with an average of $705 \pm 15$ mgC.m$^{-2}$.day$^{-1}$ (1sd, n=4) for the last 4 flux measurements that are indistinguishable from each other within $2\sigma$ (Figure 3). This observation might be explained by the fact

that over time (days to months), $CO_2$ accumulates not only in the chamber, but also in the regolith/rock connected pores surrounding the chamber in the lower compact regolith (Maquaire et al., 2002). Weathering reactions are likely to occur not only at the chamber-rock interface, but also into the rock mass over several centimetres as the weathered regolith extends to depths of up to 20 cm (Maquaire et al., 2002; Mathys and Klotz, 2008; Oostwoud Wijdenes and Ergenzinger, 1998).

    When $CO_2$ is first extracted from the chamber, the $CO_2$ stored in the surrounding pores quickly refills the chamber. It appears

that after three extractions this $CO_2$ is depleted, meaning that the more constant values correspond to the actual flux of $CO_2$ through the surface area of the chamber. We would therefore recommend that flux measurements are made on such a chamber following ~3 to 4 removals of $CO_2$, or adapted to less or more removals based on the results obtained after a series of flux measurements.

    It has to be noted that the mass of carbon ($m_C$) recovered on the zeolite molecular sieve during the period of passive trapping

($\Delta t$) cannot be directly and simply used to inform the flux of carbon through the chamber. This is because the rate of carbon trapping ($m_C/\Delta t$) follows the first Fick's law (Bertoni et al., 2004) and hence depends on the partial pressure of $CO_2$ in the chamber rather than on the flux itself. It is thus not trivial to assess the flux from the rate of carbon passive trapping ($m_C/\Delta t$)

as the flux itself may change through time. Similar reasons prevented the direct use of the amount of passively trapped $CO_2$ to estimate flux rates in previous studies (Hartley et al., 2013). Nevertheless, the rate at which $CO_2$ is trapped on the zeolite molecular sieve ($m_C/\Delta t$) is still qualitatively informative about $CO_2$ flux when compared to other sampling periods when $CO_2$ is passively trapped (see the Appendix for further information).

### 3.2 Isotope measurements and isotopic fractionation

### 3.2.1 Active sampling method

The atmospheric $CO_2$ was sampled on 27/03/2017, yielding a $\delta^{13}C$ of -9.6‰ and a $^{14}C$ activity of Fm=1.0065±0.0044. From chamber H6, we sampled $CO_2$ twice actively on 27/03/2017 (by in line trapping, Figure 3) and on 29/03/2017 (by repeated

trapping over the course of a day) both yielding ~2.1 mgC. The $^{14}C$ activities (Fm of 0.0597±0.0047 and 0.0562±0.0047, respectively) were identical within measurement uncertainty. Because the $CO_2$ originating from rock-derived organic matter and carbonate minerals is '$^{14}C$-dead', as confirmed by $^{14}C$ measurements of the organic carbon and carbonate of the rock from the studied chamber (Table 3), the atmospheric $CO_2$ input ($f_{Atm}$) can be calculated as $f_{Atm} = Fm_{Chamber}/Fm_{Atm}$. The Fm from both samples shows that only ~5.5% to 6% of the $CO_2$ trapped was of atmospheric origin and that the two active trapping

methods produce comparable results. The $\delta^{13}C$ compositions (-7.4‰ and -6.1‰, respectively) were within the range expected for a mixture of organic and inorganic carbon derived $CO_2$, but differed by ~1 ‰ for these two traps (Table 2).

It has been shown that actively trapping of $CO_2$ from headspace chambers does not induce any $\delta^{13}C$ fractionation because of near complete recovery of the $CO_2$ present in the chamber (Hardie et al., 2005). Thus the difference in the $\delta^{13}C$ composition between our two actively trapped $CO_2$ samples may reflect different relative rates of carbonate dissolution by sulfuric acid

versus organic matter oxidation over a daily timescale. Such changes in the $\delta^{13}C$ composition of the $CO_2$ sampled from field-based chambers on soils or streams have already been observed and may stem from natural environmental changes over the course of the experiments (Garnett and Hartley, 2010; Garnett et al., 2016). We cannot exclude that some diffusive processes (Davidson, 1995) within the rocks surrounding the drilled chambers or some leakage around the chamber entrance may have induced the observed 1‰ difference between our two actively trapped $CO_2$ samples. However, these samples were collected

from the exact same chamber that is likely characterized by the same diffusive processes and leakage over days. If so, the observed 1‰ is likely due to natural environmental changes in the $CO_2$ production rather than due to diffusive processes or major leaks.

### 3.2.2 Passive sampling method

From chamber H6, the $CO_2$ sample passively trapped for 101 days from mid-December 2016 to late March 2017 yielded ~11.4

mgC. The sieve cartridges have been shown to reliably trap >100 ml $CO_2$ (Garnett et al., 2009; i.e. > ~50 mgC), so the 11.4 mgC from H6 represents less than a quarter of the sieve capacity, suggesting that passive sieves can be left for at least ~6

months without becoming saturated with $CO_2$ at this field site (in reality, saturation by water vapour may be more likely to be a limiting factor). The Fm was $0.0495\pm0.0047$, which is very similar to the active trapping results, with only ~5% atmospheric $CO_2$ contamination. This is perhaps surprising since the trap was left exposed in the natural environment for three months. However, it results from the high $pCO_2$ present in the chamber throughout the time period, driving a net diffusive transfer of
$CO_2$ from chamber to the zeolite sieve. It suggests the components used to make the chamber and its linkages are not susceptible to major leaks.

The $\delta^{13}C$ composition of the passively trapped $CO_2$ was -9.4‰ and has to be corrected for a fractionation factor of $4.2 \pm 0.3$ ‰ associated with the passive trapping method (Garnett and Hardie, 2009; Garnett and Hartley, 2010) to provide the actual average $\delta^{13}C$ composition of the $CO_2$ during the duration of the experiment (here ~3 months). This fractionation is due to the
diffusive transport of $CO_2$ through air from the chamber to the zeolite molecular sieve (Davidson, 1995). The $\delta^{13}C$ composition of the passively trapped $CO_2$ sample displays a 2.0‰ and 3.3‰ depletion when compared to the $\delta^{13}C$ values obtained with the actively trapped $CO_2$ samples. This suggests that fractionation during passive trapping actually occurred, in agreement with earlier studies (Garnett et al., 2009; Garnett and Hardie, 2009; Garnett and Hartley, 2010). However, the $\delta^{13}C$ difference between actively and passively trapped $CO_2$ samples is less than the expected 4.2‰ value. It has to be noted that the passive
sampling method averages ~3 months of $CO_2$ $\delta^{13}C$ composition in the chamber, while the active sampling method averages only a few hours. Thus, the apparent "mismatch" may be due to naturally changing $CO_2$ $\delta^{13}C$ composition over time scales shorter than ~3 months and likely of the order of hours to days. This shows that both active and passive methods are complementary methods making us able to explore different timescales of sedimentary rock weathering.

### 3.2.3 The source of the $CO_2$: rock-derived organic carbon oxidation vs. carbonate dissolution by sulphuric acid

We solved the isotope-mass balance Eq. (9) for the actively trapped $CO_2$ samples from 27/03/2017 and 30/03/2017, and for the passively trapped $CO_2$ sample (Table 4). The $\delta^{13}C$ of the passively trapped $CO_2$ was corrected using the published $4.2 \pm$ 0.3 ‰ fractionation factor (Garnett and Hardie, 2009; Garnett and hartley, 2010) prior to calculations, and the $^{14}C$ activity of both the rock-derived organic carbon and carbonate end-member were set to 0, as their measured Fm were close to instrumental background (Table 3). We found very similar results for the three trapped $CO_2$ samples, yielding 5% to 6% of $CO_2$ from
atmospheric contamination, 71% to 79% of $CO_2$ from the dissolution of the carbonates by sulphuric acid and 16% to 23% of $CO_2$ from the oxidation of rock-derived organic matter (details in Table 4).

The proportion of the $CO_2$ derived from the oxidation of rock organic carbon ($f_{Rock\ OC}$) and that derived from the dissolution of carbonate by sulfuric acid ($f_{Carb}$) are corrected for the contamination of atmospheric $CO_2$ before the partitioning of the measured $CO_2$ flux. Corrected proportions ($f_x^*$, where subscript "x" is "Rock OC", or "Carb") are calculated based on the
proportions ($f_x$) found after solving the isotope-mass balance as follows:

$$f_x^* = f_x/(1 - f_{Atm}) \tag{10}$$

This shows 17% to 24% from rock-derived organic carbon and 76% to 83% from carbonate dissolution (Table 4). Therefore, for chamber H6 on 27/03/2017 for which we simultaneously measured the bulk $CO_2$ flux (705 ± 21 mgC.m$^{-2}$.day$^{-1}$), these proportions equate to a flux of 171 ± 5 mgC.m$^{-2}$.day$^{-1}$ derived from the natural oxidation of rock organic matter, and a flux of 534 ± 16 mgC.m$^{-2}$.day$^{-1}$ derived from the dissolution of carbonates by sulphuric acid (Table 4).

At the scale of chamber H6, these flux measurements imply that over a year ~109 grams of rock would be weathered by sulphuric acid to produce the carbonate-derived $CO_2$ flux (i.e., 7.1 gC produced in one year from a rock with 6.52 w% of inorganic carbon). In contrast, ~2080 grams of sedimentary rock would need to have been oxidized to produce the rock-organic carbon $CO_2$ flux (i.e., 2.3 gC produced in one year from a rock with 0.11 w% of organic carbon). The dissolution of carbonate depends on the oxidation of sulphides, and may therefore only occur locally where sulphides are concentrated. Based on these

first measurements from one chamber, the oxidation of organic carbon appears to occur more homogeneously in the rock mass.

### 3.3 First order comparison with other methods estimating $CO_2$ fluxes

To our knowledge, we report here the first attempt to directly measure the $CO_2$ flux emitted during weathering of sedimentary rocks, and trap this $CO_2$ to partition its sources using stable carbon isotopes and radiocarbon. We acknowledge that our chamber method has not been replicated, representing a limitation to our study. Nevertheless, our study is a field-based experiment,

where many environmental parameters (e.g., temperature, precipitation, water content in the unsaturated zone…) will have an impact on the weathering and erosion of the studied marls. Hence we expect that the $CO_2$ flux we measured in March 2017 and its isotopic composition will be different from measurements carried out at another times of the year. Similarly, due to the marl geochemical heterogeneities (e.g., inorganic and organic carbon contents, as well as content in sulphide mineral), the $CO_2$ flux and its isotopes could be expected to vary from one chamber to another. It is thus impossible to replicate the exact same

results we present here. We propose that future work should aim to monitor numerous chambers over seasonal changes in environmental conditions.

    While wary of these caveats, in the following sections we compare our results with other methods, to test the order of magnitudes of the $CO_2$ flux we obtained using our cylindrical chambers against previously published estimates from other regions of the world. While this exercise is challenging due to major differences in the way the $CO_2$ fluxes were estimated and

25 in the surface area and time scales (local estimates at a fixed time vs. regional estimates averaged over months/years), it is informative to assess the reliability of our method.

### 3.3.1 Rock-derived organic carbon oxidation

The flux of $CO_2$ originating from the oxidation of rock-derived organic carbon is difficult to assess. To our knowledge, there has only been one direct estimate of 0.5 gC.m$^{-2}$.yr$^{-1}$ using modelling of vadose $CO_2$ and its isotopes in Saskatchewan (Canada)

(Keller and Bacon, 1998). This is 120 times lower than our estimate in the chamber H6 of the Laval catchment (i.e., 171 ± 5 mgC.m$^{-2}$.day$^{-1}$ scaled to a year, giving 62 ± 2 gC.m$^{-2}$.yr$^{-1}$). This might be explained by the much lower erosion rates of the

Canadian site, with deep soils and stable geomorphology, compared to the Laval catchment where erosion continuously exposes rocks to oxidative weathering (Graz et al., 2012).

CO$_2$ fluxes derived from the oxidation of rock organic carbon have been indirectly estimated using geochemical proxies, such as dissolved rhenium fluxes in rivers (Dalai et al., 2002; Hilton et al., 2014; Horan et al., 2017). Our direct measurements obtained from a single chamber (H6) ($62 \pm 2$ gC.m$^{-2}$.yr$^{-1}$) are of the same order of magnitude as that calculated in highly erosive Taiwanese catchments using dissolved rhenium yields and the loss of rock organic carbon from soils (5 to 35 gC.m$^{-2}$.yr$^{-1}$) (Hilton et al., 2014; Hemingway et al., 2018). It is clearly too early to directly relate these fluxes. It is likely that individual chambers have different CO$_2$ fluxes (possibly depending on heterogeneities in the rock physical and geochemical properties, temperature, water supply to the unsaturated zone), and that CO$_2$ fluxes from a single chamber may vary throughout the year. Nevertheless, our proof of concept study suggests that direct measurements are consistent with proxy-based methods. The spatial variability in oxidation rates and its variability throughout the year are important questions which can be tested with the chamber method we describe here.

### 3.3.2 Carbonate dissolution by sulphuric acid

Inorganic carbon was the main source of the CO$_2$ flux measured during our experiment (i.e., $534 \pm 16$ mgC.m$^{-2}$.day$^{-1}$ scaled to a year, giving $195 \pm 6$ gC.m$^{-2}$.yr$^{-1}$). The dissolution of carbonate minerals by sulphuric acid (i.e., by oxidized sulphide minerals) is the simplest explanation (Calmels et al., 2007). An implication of this result is that in the Laval catchment, carbonates are weathered preferentially according to Eq. (2), i.e., releasing CO$_2$ to the atmosphere at the weathering site. This statement is supported by the average anion concentrations in the Laval stream in 2002 (Cras et al., 2007) that gives a low bicarbonate-to-sulphate ratio ($HCO_3^-/SO_4^{2-}$ ratio of ~0.35). At first order (i.e., assuming that sulphate is exclusively derived from oxidized sulphides), this observation supports the fact that carbonate weathering preferentially produces gaseous CO$_2$ (Eq. (2), i.e., $HCO_3^-/SO_4^{2-}$ ratio equal to 0) instead of dissolved inorganic carbon (Eq. (3), i.e, $HCO_3^-/SO_4^{2-}$ ratio equal to 2) at the weathering site. Because carbonate minerals are highly reactive, this means that the sulphuric acid weathering of carbonate minerals could produce a local CO$_2$ flux which starts to approach the rates of respiration in modern soils (e.g., Pirk et al. 2016).

The published river ion data can be used to estimate the weathering of carbonate minerals by sulphuric acid. From the average Ca$^{2+}$ and SO$_4^{2-}$ concentrations measured in 2002 and the average runoff (Cras et al., 2007), assuming that the weathering of carbonates produced only gaseous CO$_2$, we estimate a flux of CO$_2$ to the atmosphere of 19 to 37 gC.m$^{-2}$.yr$^{-1}$. These values could be refined by measurement of sulphur and oxygen isotopes of SO$_4^{2-}$ to partition sulphate source (Calmels et al., 2007). The river ion flux estimate is much lower than our direct measurement. This is likely due to the fact that we compare here an isolated (metre-scale) measurement to a catchment-scale average estimate which takes into account regions that have lower erosion and weathering rates. A complementary explanation would be that the flux of CO$_2$ emitted during weathering may change seasonally as a response to changes in temperature and water content in the unsaturated zone. Thus the flux we measured directly would be lower if averaged over the course of a year, hence including winter months with expected lower fluxes. This necessitates monitoring over months.

Our local direct measurement is higher than the annual flux estimate obtained for a similar highly erosive catchment in Taiwan (Liwu River) using dissolved river chemistry of ~20 $gC.m^{-2}.yr^{-1}$ (Calmels et al., 2011; Das et al., 2012; Torres et al., 2014). These values are much higher than that of less erosive major river systems like the Mackenzie River in Canada (Calmels et al., 2007; Torres et al., 2014) ($<1$ $gC.m^{-2}.yr^{-1}$) or the Ganges-Brahmaputra river system in India ($<1$ $gC.m^{-2}.yr^{-1}$) (Galy and France-Lanord, 1999; Torres et al., 2014), and supports an important control of physical erosion in the weathering of carbonates via oxidative weathering of sulphides. Our chamber-based method provides a new way to quantify this process in the field, and assess the spatial and temporal variability in $CO_2$ production by this weathering process.

**4 Conclusions**

Here, we present a reliable, innovative and relatively inexpensive way to measure the flux of $CO_2$ produced during the oxidative weathering of sedimentary rocks. The ability to trap the $CO_2$ using active or passive zeolite molecular sieves is essential, since its carbon isotopic composition ($^{12}C$, $^{13}C$, $^{14}C$) is mandatory to assess for atmospheric $CO_2$ inputs, before partitioning the $CO_2$ flux between that from oxidation of rock-derived organic carbon and carbonate dissolution by sulphuric acid. The passive method to trap the $CO_2$, i.e., leaving zeolite molecular sieve connected to a chamber for days to months, is useful to provide a time integrative sample of $CO_2$ produced during weathering. This paper is a proof of concept of the oxidative weathering of rocks: i) rock-derived organic carbon is oxidized and $CO_2$ is released directly to the atmosphere and its flux can be large enough to be directly measurable; ii) the oxidation of sulphides contained in the rocks produces sulphuric acid and dissolves carbonates and in the Laval catchment this phenomenon releases $CO_2$ directly to the atmosphere and its flux can be locally large.

**Data availability**

Raw data and flux resulting from exponential fitting of data are available in the supplementary material.

**Appendix**

Here we explain how the mass of $CO_2$ accumulated on the passive traps over several months may be compared to short-term flux measurements made during the active trapping method. Passive sampling is a practical application of Fick's first law (Bertoni et al., 2004). In our case it is related to the diffusion (D) of $CO_2$ molecules in air caused by the gradient of $CO_2$ partial pressure between that of the chamber ($pCO_{2,Ch}$) and that of the zeolite trap ($pCO_{2,zeolite}$). This diffusion is defined for a length of time ($\Delta t$) and is limited to the internal side of the tube linking the chamber to the zeolite trap, i.e. the diffusion path characterized by the tube length (L) and tube section area (a). It results in the trapping of a certain mass of carbon ($m_C$) in the zeolite trap. In this case, first Fick's law may be written as follows:

$$pCO_{2,Ch} - pCO_{2,zeolite} = \frac{m_C}{\Delta t} \frac{L}{aD} \frac{RT}{PM_C} 10^6 \tag{A-1}$$

R is the gas constant, T is temperature, P is pressure and $M_C$ is the molar mass of carbon. Factor $10^6 \times RT/PM_C$ converts grams of carbon to $cm^3$ of $CO_2$, and $pCO_2$ is here in ppm ($cm^3/m^3$). Note that the $pCO_{2,zeolite}$ in the zeolite trap is equal to 0 ppm, since the zeolite is the $CO_2$ absorber. The equation thus reduces to:

$$pCO_{2,Ch} = \frac{m_C}{\Delta t} \frac{L}{aD} \frac{RT}{PM_C} 10^6 \tag{A-2}$$

Equation (A-2) can be used reconstruct the average partial pressure of $CO_2$ in the chamber $pCO_{2,Ch}$ during the sampling duration ($\Delta t$). Eq. A-2 also indicates that the passive trapping is only directly linked to the partial pressure in the chamber over the sampling length of time $\Delta t$. In other words, passive sampling is not related in a simple way to the flux of $CO_2$ through the chamber.

In order to relate long-term passive sampling to short-term $CO_2$ flux measurements, we assume that the evolution of the $pCO_2$ in the chamber can be described following an exponential law (Pirk et al., 2016; see Eq. 7-8 of main text), and we can describe the $pCO_{2,Ch}$ in the chamber based on other parameters:

$$pCO_{2,Ch} = \frac{1}{V_{Ch}} \left[ \frac{q}{\lambda} \left( 1 - \exp(-\lambda \Delta t) \right) + m_0 \right] \frac{RT}{PM_C} 10^6 \tag{A-3}$$

$V_{Ch}$ is the volume of the chamber, q is the initial rate of carbon accumulation in the chamber, $m_0$ is the initial mass of carbon
in the chamber (a value that corresponds to 400ppm of $CO_2$ in the volume of the chamber). $\lambda$, per unit of time, is the parameter that describes the diffusive processes responsible for the non-linear accumulation of carbon in the chamber. In the case of long-term passive sampling $\Delta t$ is very large (~3 months and thus ~150,000 minutes). Thus $\exp(-\lambda \Delta t)$ tends to 0 and Eq. A-3 simplifies to:

$$pCO_{2,Ch} = \frac{1}{V_{Ch}} \left[ \frac{q}{\lambda} + m_0 \right] \frac{RT}{PM_C} 10^6 \tag{A-4}$$

Note that Eq. A-4 can be written only if we assume that the initial rate of carbon accumulating in the chamber (q) does not change over time. This assumption may be violated because q is unlikely to stay constant over time for various reasons including natural variability in $CO_2$ production and also changes in the diffusive processes (parameter $\lambda$) when $pCO_2$ builds up in the chamber. Equating Eq. A-2 and Eq. A-4, we obtain:

$$\frac{m_C}{\Delta t} \frac{L}{aD} \frac{RT}{PM_C} 10^6 = \frac{1}{V_{Ch}} \left[ \frac{q}{\lambda} + m_0 \right] \frac{RT}{PM_C} 10^6 \tag{A-5}$$

Hence we can derive the rate at which carbon accumulates into the chamber based on the passive trapping parameters and $\lambda$, which is measured in the field over short time periods (i.e., during the short-term flux measurements when $CO_2$ is actively actively trapped – see Eq. 7-8 in the main text):

$$q = \lambda \left( \frac{m_C}{\Delta t} \frac{L}{aD} V_{Ch} - m_0 \right) \tag{A-6}$$

The flux can be inferred from the later equation using the internal surface area of the chamber (S). If q was in mgC/min, then the flux of carbon Q in $mgC/m^2/day$ is:

$$Q = \lambda \left( \frac{m_C}{\Delta t} \frac{L}{aD} V_{Ch} - m_0 \right) 1440/S \tag{A-7}$$

10   We can determine most of the parameters of Eq. A-6 independently from the short-term flux (Q or q) measurements, except for $\lambda$. For instance $m_C$, $m_0$, $V_{Ch}$, S, $\Delta t$, a and L can be measured and D (diffusion of $CO_2$ in air) can be inferred from the literature. However, $\lambda$ is determined using the short-term flux measurements, along with the flux (i.e., Q or q). Thus estimating the flux of $CO_2$ based on the rate of carbon passively trapped in the zeolite trap ($m_C/\Delta t$) is not independent from the short-term $CO_2$ flux measurements. Hence, comparing a $CO_2$ flux inferred from the mass of carbon $m_C$ recovered using the passive trap and calculated using Eq. A-6 and Eq. A-7, and the $CO_2$ flux actually measured during our short-term experiments, is somewhat 15   circular because they are not determined independently from each other.

For longer monitoring of field work sites, the mass of carbon trapped is still qualitatively informative. This is because $m_C$ and the rate carbon trapping per unit of time ($m_C/\Delta t$) are proportional to the flux of carbon Q to the chamber and parameter $\lambda$. This is illustrated easily by writing equation Eq. A-7 differently:

$$m_C/\Delta t \propto Q/\lambda \tag{A-8}$$

20   where the left-hand part of Eq. 8 are the parameters measured during passive trapping and the right-hand part of Eq. A-8 are the parameters measured during short-term flux measurements. Interpretations of changes in $m_C/\Delta t$ thus give qualitative constraint on $CO_2$ fluxes over time. Future work might investigate whether the parameter $\lambda$ can be characterised for a chamber independently from the active $CO_2$ flux measurements. If it can, the passive trap method can be used not only qualitatively (e.g. to look for changes in the mass of $CO_2$ collected on passive traps through time), but quantitatively (i.e. the monthly time-25   integrated $CO_2$ flux).

## Author contribution

RGH conceived the research and designed the study with GS and MHG. GS, RGH and SK carried out chamber installation. GS carried out flux measurements and sample collection with on-field assistance of RGH, MD and MO. GS, RGH, MHG and TC analysed the data. GS, RGH and MHG interpreted the data. GS wrote the manuscript with inputs from RGH and MHG. All co-authors commented on the manuscript.

## Acknowledgements

This research was funded by a European Research Council Starting Grant to Robert G. Hilton (ROC-CO$_2$ project, grant 678779) and by the Natural Environment Research Council (NERC) Radiocarbon Facility (East Kilbride). We thank staff at NERC RCF and SUERC. We thank Anne-Eléonore Paquier and Clément Flaux for assistance on field in December 2016. We thank Jérôme Gaillardet and Caroline Le Bouteiller for collaborative access to the infrastructure at the Draix Bléone Observatory, (IRSTEA). Camille Cros and an anonymous reviewer are thanked for their comments which improved the manuscript.

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

**Tables**

**Table 1. Dimensions of a typical chamber[a, b]**

| Inner diameter | Depth | PVC tubing | Depth of insertion of PVC tubing | Depth of insertion of rubber stopper | Chamber volume | Area of exchange with surrounding rock |
|---|---|---|---|---|---|---|
| cm | cm | cm | cm | cm | cm$^3$ | cm$^2$ |
| 2.9 (2.8 - 3.0) | 41 (40.5 - 41.5) | 4 (3.5 - 4.5) | 1.5 (1 - 2) | 1 (0.75 - 1.25) | 281 (252 - 312) | 366 (345 - 389) |

[a]Chamber H6 drilled on 13/12/2016 in the catchment of the Laval stream (Draix, France; N44.14061°, E06.36289°)

[b]Given as ranges: median (min - max)

**Table 2. Isotopic composition of the CO$_2$ sampled with the zeolite molecular sieves**

| Sample label | Publication number | Method | Mass of carbon sampled (mg) | $\delta^{13}C$ (‰VPDB) | Fraction modern |
|---|---|---|---|---|---|
| DRA16-H6-1512-P | SUERC-73091 | Passive[a] | 11.4 | -9.4 | 0.0495 ± 0.0047 |
| DRA17-H6-2803-A | SUERC-73096 | Active | 2.1 | -7.4 | 0.0597 ± 0.0047 |
| DRA17-H6-3003-A | SUERC-73094 | Active | 2.1 | -6.1 | 0.0562 ± 0.0047 |
| DRA17-ATM-2703 | SUERC-73095 | Active | 3.8 | -9.6 | 1.0065 ± 0.0044 |

[a] sampled passively for 100.84 days

**Table 3. Geochemical compositions of the end-members involved in the isotopic mass balance (Eq. 9) were measured from the rock sampled during the drilling of chamber H6[a], and from an atmospheric CO$_2$ sampled actively with a zeolite molecular sieve (Table 2).**

| | Content (weight %) | Publication number | $\delta^{13}C$ (‰VPDB) | Fraction modern |
|---|---|---|---|---|
| Total Inorganic Carbon | 6.52±0.6 (n=3) | SUERC-74506 | 0.3 ± 0.1 | 0.0032 ± 0.0006 |
| Total Organic Carbon | 0.11±0.7 (n=3) | UCIAMS-192874 | -30.8 ± 0.1 | 0.0125 ± 0.0039 |
| Atmospheric CO$_2$ | n/a | SUERC-73095 | -9.6 ± 0.1 | 1.0065 ± 0.0044 |

[a]in house label of this sample was DRA16-78

**Table 4. Isotope-mass balance results**

| Sample label | Publication number | Method | Sources | Proportion (%) | Proportion corrected for atmospheric contribution (%) | Partitioned flux ($mgC.m^{-2}.day^{-1}$) |
|---|---|---|---|---|---|---|
| DRA16-H6-1512-P | SUERC-73091 | Passive[a] | Atmosphere | $4.9 \pm 0.5$ | – | – |
| | | | Carbonates | $79.0 \pm 1.1$ | $83.0 \pm 1.1$ | – |
| | | | Rock Organic Carbon | $16.1 \pm 1.0$ | $17.0 \pm 1.1$ | – |
| DRA17-H6-2803-A | SUERC-73096 | Active | Atmosphere | $5.9 \pm 0.5$ | – | – |
| | | | Carbonates | $71.2 \pm 0.5$ | $75.7 \pm 0.4$ | $534 \pm 16$[b] |
| | | | Rock Organic Carbon | $22.9 \pm 0.4$ | $24.3 \pm 0.4$ | $171 \pm 5$[b] |
| DRA17-H6-3003-A | SUERC-73094 | Active | Atmosphere | $5.6 \pm 0.5$ | – | – |
| | | | Carbonates | $75.6 \pm 0.5$ | $80.1 \pm 0.4$ | – |
| | | | Rock Organic Carbon | $18.8 \pm 0.4$ | $19.9 \pm 0.4$ | – |

[a] before solving isotope-mass balance, the $\delta^{13}C$ of the passive sample was corrected for a fractionation factor of $4.2 \pm 0.3$ ‰ (Garnett and Hardie, 2009; Garnett and Hartley, 2010)

[b] from a measured bulk $CO_2$ flux of $705 \pm 21$ $mgC.m^{-2}.day^{-1}$

**Figures**

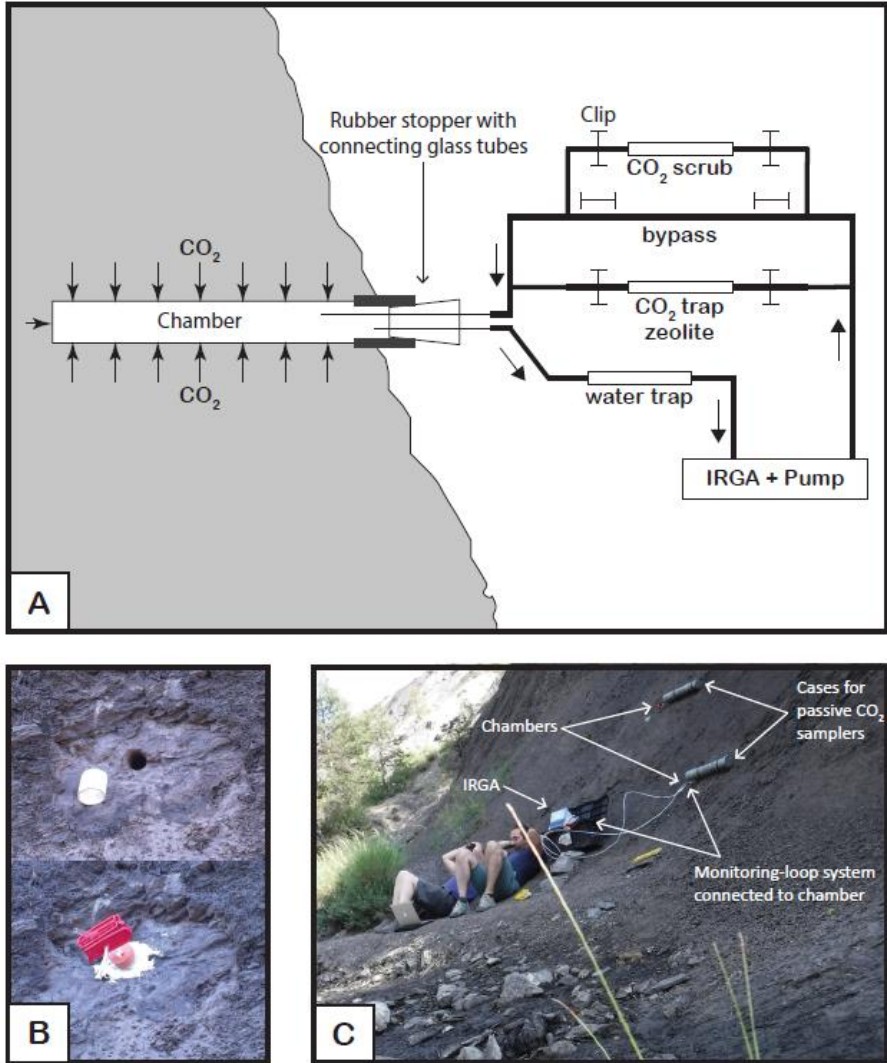

Figure 1: A: Schematic diagram of the closed-loop monitoring-sampling system connected to the natural weathering chamber. Gas flow pathways are controlled by opening and closing the clips. Clips removed from the bypass allow $pCO_2$ in the chamber to be monitored (IRGA stands for Infra-Red Gas Analyzer), thus measuring $CO_2$ flux and ensuring that enough $CO_2$ accumulated in the chamber for [14]C analysis. To remove $CO_2$ from the line before connecting to the chamber, clips are moved from the $CO_2$ scrub (soda lime). When connected to the chamber, the $CO_2$ scrub can be used to lower the $CO_2$ concentration before flux measurement. To collect $CO_2$ in the chamber for isotope analyses, clips are removed from the zeolite molecular sieve cartridge. B: Pictures showing the chamber design. Top picture is chamber (H6), diameter 2.9cm, drilled in the rock on a cleared surface, with white PVC tubing inserted at the outlet. Bottom picture shows the rubber stopper fitted in the PVC tubing. Two glass tubes go through the rubber stopper and are fitted with Tygon tubing, sealed with the red clips, and the exterior of the chamber is sealed with outdoor sealant. C: View of the field site showing two chambers (top chamber is H6 and lower chamber is H4). The lower chamber is connected to the closed-loop system and is being monitored for flux measurement. The two large grey PVC tubes attached to the rock on the right of the chambers are cases in which zeolite molecular sieves are installed and left for months when connected to the chamber for passive $CO_2$ trapping.

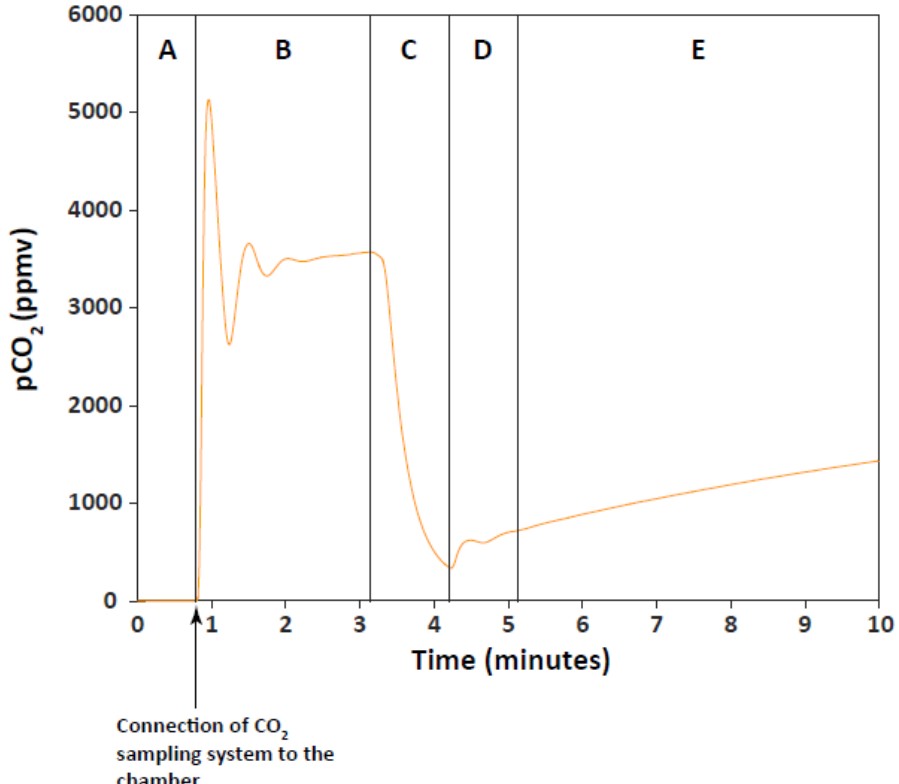

**Figure 2: An example of the monitoring of the CO₂ accumulating in a chamber. The orange curve is the partial pressure of CO₂ (pCO₂, in parts per million volume) through time in chamber H6 on 27/03/2017. A: The CO₂ sampling-monitoring system is not connected to the chamber. Atmospheric CO₂ has been removed from the system (pCO₂ = 0 ppm) using the CO₂ scrub cartridge. B:**
5    **The closed-loop monitoring system has been connected to the chamber. pCO₂ increases to reach a maximum value of ~5100 ppm, then drops and equilibrates to ~3500ppm. This pattern reflects the increase in the total volume (by the volume of the CO₂ sampling-monitoring system) which decreases pCO₂ and requires some time for the pCO₂ to equilibrate. We determined that when connected to the chamber, the maximum value of pCO₂ read is 0.94 the actual pCO₂ in the chamber. C: The CO₂ in the chamber is lowered (scrubbed with the CO₂ scrub, or trapped with the zeolite molecular sieve) to near atmospheric pCO₂. D: residual CO₂ that in the**
10    **chamber homogenized with the rest of the total volume "artificially" increasing pCO₂ quickly. E: pCO₂ in chamber is monitored, reflecting the flux of CO₂ from the rock surface to the chamber.**

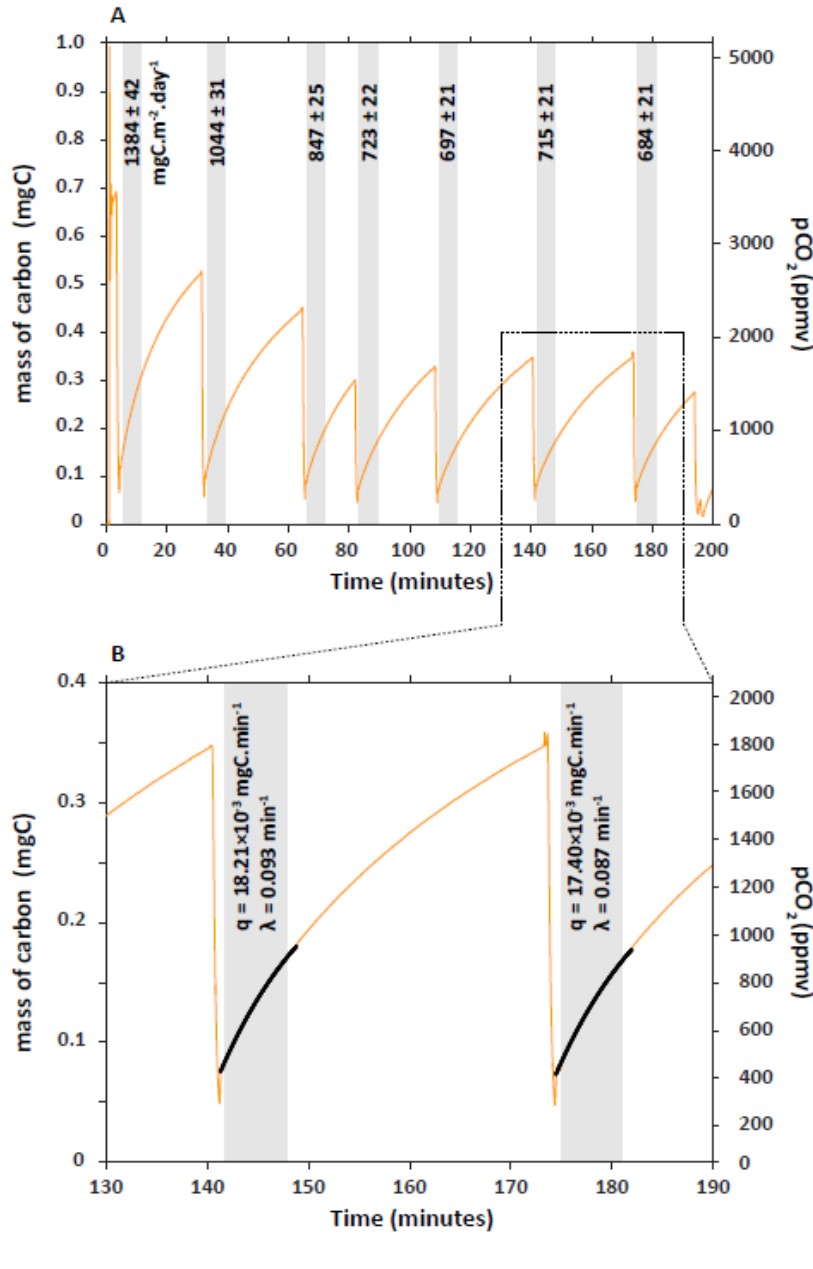

**Figure 3**

**Figure 3: A: Series of carbon flux measurements for chamber H6 on 27/03/2017. $CO_2$ concentration ($pCO_2$) was converted into mass of carbon (mgC) following Eq. (4) and Eq. (5). Flux of $CO_2$ – the numbers associated to shaded boxes – are given in $mgC.m^{-2}.day^{-1}$. B: Close–up of how fluxes were calculated from the rate of carbon accumulation (parameter q) by fitting the exponential model described in Eq. (7) and (8) for 6 minutes (shaded box).**

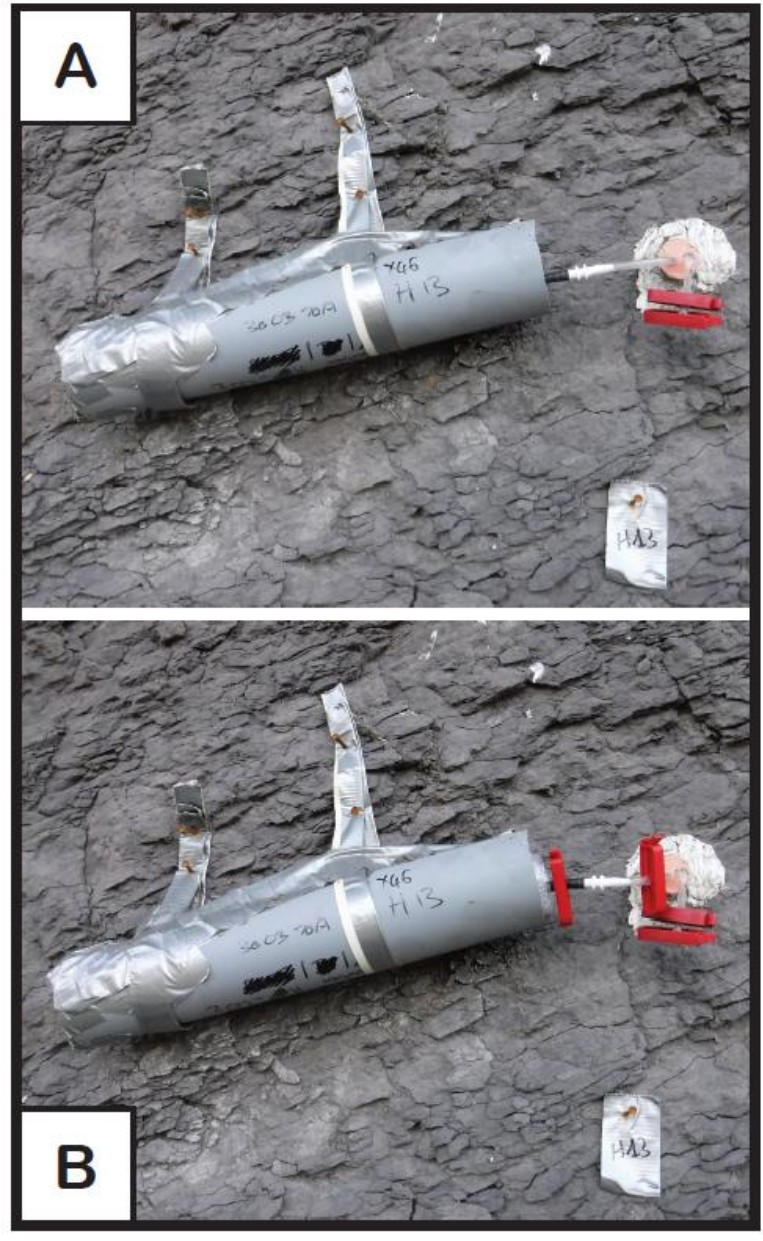

**Figure** 4**: A: Zeolite molecular sieve connected to a chamber for passive CO₂ trapping. The zeolite molecular sieve is encased in the grey PVC tubing and connected for months to the chamber using a white connector. B: Zeolite molecular sieve ready to be disconnected from chamber. The red clips are positioned so that they seal both the zeolite molecular sieve and the chamber.**