# Peer review of "Technical note: *in situ* measurement of flux and isotopic composition of CO2 released during oxidative weathering of sedimentary rocks"

_Biogeosciences, 2017_

## Referee Comment (RC1) · C. Cros (Referee) · 30 Jan 2018

The main objective of this study is to present the results from a new method estimating the amount and origins of CO2 released during the weathering of sedimentary rocks. The released CO2 might have two origins: the degradation of limestone by sulphuric acid and the (geo)respiration of organic carbon. The method is based on a respiration chamber drilled directly on the rock. The released CO2 is measured by following the short-term (hours) accumulation of CO2 in the respiration chamber after lowering the pCO2 to near atmospheric pC02. Three sources of CO2 could contribute to the accumulation of CO2 in the chamber: the atmospheric CO2 due to possible contaminations and/or leaks, carbonates and organic C. These tree sources were successfully separated by analyzing the 12C, 13C and 14C isotopic composition of released CO2. To this end, the CO2 was trapped by two systems of zeolite that actively (hours) or passively (months) trap the released CO2. The study provides first estimations of CO2 emissions by oxidative weathering of sedimentary rocks, which were unfortunately not compared to another method. I would say this is the main limit of this study. The isotopic analysis of CO2 provided evidence of low contamination of sampled CO2 by atmospheric CO2, which validates the tightness of respiration chamber and method of CO2 sampling. It also allowed providing first estimation of contribution of the two mechanisms of weathering contributing to CO2 emissions (acid degradation of limestone versus organic C respiration), which could not be compared to another method. Although the main results could not be validated by using another independent method, these first estimations are useful and timely justifying a possible publication of this manuscript in BG. However, there are major drawbacks that deserve major revisions of the manuscript. First of all, the authors should acknowledge the fact that their method is not compared to other ones and is not replicated strongly limits conclusions about the accuracy, sensitivity and reproducibility of such method. Why did you not compare the results obtained by several respiration chambers? Why did not compare your estimation of released CO2 based on short term measurement of CO2 accumulation in chamber with the amount of CO2 you trapped after several months of passive CO2 trapping by zeolite. This comparison could be a first way to evaluate your method of estimation. The method of partitioning of CO2 sources is absolutely not well introduced and explained. For example, P5 L3-6 you should explain that these three sources of CO2 (atmospheric, limestone, organic C) have different isotopic composition. And you should give some order of magnitude maybe. The system of equations (9) should be carefully explained, in particular all the variables must be defined (what does Fm mean?). Table 3 must include the isotopic composition of air of the site. Results and discussions must be presented in distinct sections to clearly separate facts from their interpretations (and fit to the standard of BG). To my opinion, the estimation of released

CO2 by short-term measurement should be compared to the amount of CO2 trapped by zeolite. If this is not comparable, I expect detailed explanations of reasons. Title of section 3.3 is not clear and does not reflect the content. It seems that the objective of this section is to make a first comparison of estimation made by this study with published results from other sites. If I understand well, your estimated amounts are far above the ones present in the literature. You should give some interpretations of these differences including the fact that your method has some bias that could lead to overestimations. First, the drilling can generate hyperactive surface by providing dust (small particules with high surface areas). Second, fresh surface is rich in organic C and limestone (because not previously explosed to O2). The CO2 may diffuse from pores of surrounding rock to chamber signifying that the surface of rock contributing to these estimations is larger than the sole surface of chamber. Detailed comments P3L11: why did you set up these two methods of trapping? The idea must be introduced before. P4 L4. The rock-drill was used to dig a hole or a cylinder. This is only a part of the chamber. P4L24-27 The drilling makes powder that can stay on the surface. I guess that dust is highly reactive compared to rock that, has already been exposed to weather and oxidation since many years. This should be stated and discussed somewhere, maybe in the discussion section. P6 L11-14 This text has no meaning for me, could you try to better explain? Concerning this section on the estimation of CO2 release, how did you manage the fact that released CO2 can accumulate in water present in the rock under the form of carbonates? Equations 9. You must say that this system of three equations can calculate three unknowns: atmopsheric CO2, limestone originating CO2, organic C originating CO2. Define all variables. Results and discussions must be separated. That will clarify your results and explanations. P8 L18-20. Not necessary, the dissolution of CO2 in water and formation of carbonates could lead to a non linear response. P9 L1-2 This decrease could result from an exhaustion of CO2 of pores surrounding the chambers (at the beginning of measurement these pores contribute much to the accumulation of CO2 in the chamber and they become empty with time). P9L17 I disagree. The amount of atmospheric CO2 is given by your system of equations (9) P9L20

Cite Table 4 P10 L10-11 You should better explain why do you make a correction for atmospheric CO2 although this contribution was already considered in your system of equations? After reading Table 4, I understood but you should better explain in the text. P10 L21-22 Change titles, they are not helpful for the understanding. P11 L3 You should add text to explain that your method must be compared with other (direct or indirect) methods on the same site P11 L5-9. The logical link between these sentences is not obvious. P11L16 the numbers "19 to 37 gC m-2 yr-1" must be compared to "206 gC m-2 yr-1" of your study? The difference is enormous and deserves some explanations. Figure 1: I do not see what pictures B and C bring to the story. Have you checked that all the materials you use, especially the products used to seal and make tight (e.g. expansive foam etc), does not emit CO2?

---

## Referee Comment (RC2) · Anonymous Referee #2 · 3 Apr 2018

In their Technical Note entitled, "*in-situ* measurement of flux and isotopic composition of CO$_2$ released during oxidative weathering of sedimentary rocks", Soulet et al. report the results of a "proof-of-concept" study aimed at determining the release rate of carbon dioxide from outcrops of weathered shales and partitioning this carbon dioxide between inorganic and organic sources using C isotopes. The authors clearly describe the design, implementation, and data analysis for their rock weathering chambers in such a way that I am confident that I, or any other researcher, could implement this technique elsewhere. While I ultimately think that this paper should be published in *Biogeosciences*, I have a few comments that I would like to see the authors address (described below).

[Figure]

Firstly, I am confused by the distinction between a "direct" and "indirect" measure of a chemical weathering reaction. On Page 1 line 12, the authors imply that tracking reaction products (e.g., dissolved sulfate in rivers) is an indirect method. However, as carbon dioxide is also a reaction product, I do not see how their method is any more direct than measuring sulfate concentrations. Moreover, the relationship between the amount of product consumed (carbonate or organic carbon) and the amount of carbon dioxide release can be strongly modulated by the buffering capacity of natural waters. As a result, tracking carbon dioxide release may lead to a different assessment of the extent of reaction relative to a product that doesn't partition into both the fluid and gas phase (e.g., sulfate ion). That being said, I do agree that their method provides a different perspective on weathering reactions than measuring the dissolved or solid phase chemistry of rivers. In particular, I think the Soulet et al. method averages over very different temporal and spatial scales (see below) that make it a nice complement to river based approaches. Perhaps some more clarity as to what the authors mean by direct versus indirect would be helpful.

The different stoichiometries for carbonate weathering by sulfuric acid (CWSA) presented as equations 2 and 3 have appeared elsewhere in the literature. However, I am not convinced that, in the context of this paper, there is a real distinction that can be made. The dissolution of one mole of calcium carbonate releases one mole of calcium ion ($Ca^{2+}$) and one mole of dissolved inorganic carbon (DIC; $2H^+ + CaCO_3 \rightarrow Ca^{2+} + H_2CO_3$), which is equivalent to 2 units of alkalinity per unit of DIC. The generation of sulfuric acid from pyrite oxidation can titrate these 2 units of alkalinity leading to a net reaction for CWSA that results in 0 units of alkalinity generation per unit of DIC generation ($H_2SO_4 + CaCO_3 \rightarrow Ca^{2+} + SO_4^{2-} + H_2CO_3$; equivalent to Equation 2 of Soulet et al.). Equation 3 of Soulet et al. predicts 1 unit of alkalinity generation per unit of DIC generation. In this way, it can be viewed as a 50/50 mixture of carbonate weathering by carbonic and sulfuric acids instead of a distinct pathway for CWSA.

Furthermore, the idea that Equation 2 reflects an "immediate" release of carbon dioxide to the atmosphere misses the fact that the aqueous chemistry of weathering fluids will strongly modulate this flux. If there is sufficient generation of alkalinity from silicate weathering, the carbon dioxide produced from CWSA will partition more into the dissolved phase despite generally following the stoichiometry of Equation 2. Similarly, springs developed in carbonate terrains that lack abundant pyrite still degass carbon dioxide into the atmosphere despite the fact that the reaction for carbonate weathering by carbonic acid is often written as generating bicarbonate ion. In other words, without more constraints on the fluid composition, it difficult to directly relate the extent of an individual weathering reaction to changes in carbon dioxide concentrations (e.g., see Soetart et al. 2007 Maine Chemistry).

In general, this study lacks replication. While I do not think that this is a critical issue, it'd be worth acknowledging some of the limitations and/or adding more analysis where possible. For example, two chambers are shown in Figure 1C. Is there not two chamber's worth of data to show? Similarly, I am not sure if I found at what depth below the land surface the chamber was placed. Presumably this depth will have a large effect on the results. What depth was selected and why?

Page 2 Line 24 - There are many other papers that have used S (and O) isotope ratios to partition the sulfate budget including some that precede the Calmels et al. 2007 paper. For example:

* Cameron, Eion M., et al. "Isotopic and elemental hydrogeochemistry of a major river system: Fraser River, British Columbia, Canada." Chemical geology 122.1-4 (1995): 149-169.

* Spence, Jody, and Kevin Telmer. "The role of sulfur in chemical weathering and atmospheric $CO_2$ fluxes: evidence from major ions, $\delta$13CDIC, and $\delta$34SSO4 in rivers of the Canadian Cordillera." Geochimica et Cosmochimica Acta 69.23 (2005): 5441-5458.

* Das, Anirban, Chuan-Hsiung Chung, and Chen-Feng You. "Disproportionately high

rates of sulfide oxidation from mountainous river basins of Taiwan orogeny: Sulfur isotope evidence." Geophysical Research Letters 39.12 (2012).

* Turchyn, Alexandra V., et al. "Isotope evidence for secondary sulfide precipitation along the Marsyandi River, Nepal, Himalayas." Earth and Planetary Science Letters 374 (2013): 36-46.

* Hindshaw, Ruth S., et al. "Influence of glaciation on mechanisms of mineral weathering in two high Arctic catchments." Chemical Geology 420 (2016): 37-50.

* Torres, Mark A., et al. "The acid and alkalinity budgets of weathering in the Andes–Amazon system: Insights into the erosional control of global biogeochemical cycles." Earth and Planetary Science Letters 450 (2016): 381-391.

Page 6 Line 15 - I'd appreciate a few sentences that explain directly how $V_{ch}$ and S were determined. I assume that the dimensions of the drill hole and the assumption that it was shaped as a perfect cylinder were used. However, this ignores the fact the the chamber walls are rough and not perfectly impervious. As a result, you are likely to get carbon dioxide from pores and cracks that intersect the chamber walls as alluded to on Page 9 Line 6. I'd appreciate some additional discussion on how this effects area-normalized estimates of carbon dioxide production rates.

Furthermore, are their constraints from porosity, permeability, grain-size, and/or fracture density measurements that can inform the "effective" volume that the chamber samples? Or, could the mass of $CO_2$ removed during the first few flushes inform this volume? Being able to determine an "effective" volume (as controlled by porosity, permeability, fracture density, grain size, etc.) would help others trying to replicate the methodology in determining if a site would be appropriate based on rock properties.

Page 8 Line 27 - How realistic is it that the chamber has such a high $pO_2$? My understanding of evidence from the oxygen isotopic composition of sulfate (e.g., Calmels et al. 2007), pyrite reactions fronts (Brantley et al. 2013 ESPL), and gas chemistry in

wells (Kim et al. 2014 GCA 2017 GCA) is that oxidative weathering takes place under relatively low $pO_2$ conditions for many systems. Does this mean that your method provides a maximum estimate of reaction rates?

Page 9 Line 9 - For the analysis of $CO_2$ fluxes, it is stated that 3-4 flushes are necessary to get the "true" flux determination. What statistical criteria was this determination based on? Similarly, what is the basis for designating 6 minutes as the amount of time to fit the carbon dioxide accumulation curve (Page 6 Line 11)?. How do the calculated averages and standard deviations of $CO_2$ flux vary with measurement / integration time?

Page 9 Line 20 - I am not convinced that the difference between the 2 carbon isotopic samples reflects process and not fractionation. The analysis of carbon dioxide fluxes explicitly assumes that there are leaks in the system, which may induce fractionation. Similarly, two different methods were used for these samples. Finally, if the balance between oxidative reactions can vary daily, then why is the entire difference in the isotopic composition of $CO_2$ derived from the passive trapping method assumed to result from fractionation. In general, a better discussion of which isotopic signals are attributed to environmental process vs. sampling-induced fractionation and why would be helpful.

Page 10 Line 15 - **This is very interesting!**. In other words, the release ratio of inorganic to organic carbon determined by carbon isotope ratios is different than the relative abundances of inorganic and organic carbon present in the underlying rock. Specifically, the isotopic method "sees" more organic carbon than would be expected if one "unit" of rock was congruently weathered. Does this make sense with what is known about carbon and sulfur reaction fronts in weathering profiles?

Page 10 Line 21 - While I understand the motivation behind including section 3.3, I think that the different area normalization schemes between the chambers and river-based measurements precludes direct comparison. The area normalization in river systems

refers to the catchment area. However, weathering takes place at depth within porous media such that the true surface area of reactive material that rivers source solutes from is likely poorly approximated by the catchment surface area. In the chamber experiments, the area normalization refers to the surface area of the chamber walls, which likely more closely approximates the true "reactive" surface area (see above). At the very least, this discrepancy between area normalization schemes should be discussed before generating comparisons between the different datasets. Depending on how reactive surface area scales with catchment area, the fact that the chamber-based estimates are close in magnitude to the river-based estimates may actually mean that there is a large discrepancy in the rates that they predict.

---

## Author Response (AR1)

Dr. Guillaume Soulet Post-Doctoral Research Associate

Department of Geography Durham University South Road, Durham, DH1 3LE, UK

*Office*: +44 (0)191 33 41801 *Email*: guillaume.s.soulet@durham.ac.uk

The Editor Biogeosciences journal

May, 25th, 2018

Dear Editor Sébastien Fontaine,

Thank you for handling our manuscript. We have carefully considered and addressed both Reviewers suggestions and comments. These have resulted in revisions throughout the manuscript. In summary here, the main changes are:

- As prompted by Reviewer 1, and as you suggested, we have added the mathematical description of how longer-term passive trapping of CO2 onto zeolite sieves can be related to shorter-term active measurements of CO2 emissions. This text is based on our reply to reviews as published in *Biogeosciences Discussions*. We provide an overview of this in revised manuscript main text (Section 3.1) and the full details as an Appendix.
- 2) We now flag that the method we present is one approach to do this (final paragraph of Section 1), that it has not been replicated here and explain the challenges of doing so, before expanding on the resulting caveats (new paragraphs at the start of Section 3.3). These revisions address comments raised by both Reviewers as highlighted in your comments.
- 3) We now provide the CO2 flux measurements in units of mgC m-2 day-1 at the first instance. This is intended to clarify to the reader that our fluxes represent those over a short period of time, allowing us to be more cautious when we compare these first CO2 emission measurements to published work on oxidative weathering fluxes.
- 4) We provide an expanded discussion of the characteristics of the weathering zone and regolith production. We are more careful in our use of the 'direct' and 'indirect' terminology when referring to measurements of weathering fluxes (Reviewer 2).

We have upload our revised manuscript and we provide a word track-changes version here, so you can follow the nature and extent of the revisions.

Best regards,

fuilanne Jouh

Dr. Guillaume Soulet on behalf of all co-authors

Soulet et al. response to Reviewer 1, for "Technical Note: *in situ* measurement of flux and isotopic composition of  $CO_2$  released during oxidative weathering of sedimentary rocks"

**We thank Camille Cros for reviewing our manuscript and for her helpful feedback. Below we address these comments, along with corresponding changes made to the manuscript text.**

The main objective of this study is to present the results from a new method estimating the amount and origins of CO2 released during the weathering of sedimentary rocks. The released CO2 might have two origins: the degradation of limestone by sulphuric acid and the (geo)respiration of organic carbon. The method is based on a respiration chamber drilled directly on the rock. The released CO2 is measured by following the short-term (hours) accumulation of CO2 in the respiration chamber after lowering the pCO2 to near atmospheric pCO2. Three sources of CO2 could contribute to the accumulation of CO2 in the chamber: the atmospheric CO2 due to possible contaminations and/or leaks, carbonates and organic C. These tree sources were successfully separated by analyzing the 12C, 13C and 14C isotopic composition of released CO2. To this end, the CO2 was trapped by two systems of zeolite that actively (hours) or passively (months) trap the released CO2.

The study provides first estimations of CO2 emissions by oxidative weathering of sedimentary rocks, which were unfortunately not compared to another method. I would say this is the main limit of this study. The isotopic analysis of CO2 provided evidence of low contamination of sampled CO2 by atmospheric CO2, which validates the tightness of respiration chamber and method of CO2 sampling. It also allowed providing first estimation of contribution of the two mechanisms of weathering contributing to CO2 emissions (acid degradation of limestone versus organic C respiration), which could not be compared to another method. Although the main results could not be validated by using another independent method, these first estimations are useful and timely justifying a possible publication of this manuscript in BG. However, there are major drawbacks that deserve major revisions of the manuscript. First of all, the authors should acknowledge the fact that their method is not compared to other ones and is not replicated strongly limits conclusions about the accuracy, sensitivity and reproducibility of such method.

RE: A main concern of the reviewer is the lack of replication and comparison with other methods. To our knowledge, our study is the very first attempt to detect, measure, quantify, trap and partition the source of  $CO_2$  emissions during weathering of sedimentary rocks in such settings where weathering and erosion rates are high. It is therefore not possible to compare our results to others. That is why, in the discussion section, we attempt to relate our results to other studies that have estimated the  $CO_2$  flux at the scale of river catchments using geochemical proxies in river water. The fluxes are of the same order of magnitude, yet, they are obviously different. We provided some explanations for the discrepancies in the original manuscript.

In terms of replication, this is challenging. Our measurements are not set up in a laboratory where parameters could be controlled. Instead we are working directly in the field where environmental parameters vary with time and space. For example, a single chamber may be expected to provide different fluxes through time as a response to seasonal environmental changes (temperature, humidity). If one were to compare two different chambers, they are likely to yield different results at the same time because of local differences in the weathering substrate (caused by differences in the chemistry of the weathered rock, fracturing, connectivity, porosity, slope of the rock face...). Therefore, to take this method further, we would recommend a field set up which allows for repeated measurements over seasons, and one with multiple chambers to examine the importance of weathering substrate.

Accordingly, to acknowledge that one method is outlined in our manuscript (with active and passive  $CO_2$  approaches), and so there is no comparison to an alternative method, we provide a summary of the above discussion in the revised version (revised manuscript section 3.3 P12 L15-27).

**Why did you not compare the results obtained by several respiration chambers?**

RE: Our Technical Note is a "proof-of-concept" study, of which the goal is to test and discuss the feasibility of directly measuring CO2 emissions in sedimentary rocks during weathering and our ability to trap this CO2 to

measure its isotopic composition and determine its source. We demonstrate here it is possible, and the results from only one chamber are needed for this purpose. Comparing the results obtained from several chambers would be very interesting but it is the focus of a very different study which, for instance, would aim to discuss the variability over space and time of the CO2 emissions and source proportions.

Why did not compare your estimation of released CO2 based on short term measurement of CO2 accumulation in chamber with the amount of CO2 you trapped after several months of passive CO2 trapping by zeolite. This comparison could be a first way to evaluate your method of estimation. RE: See detailed answer below.

The method of partitioning of CO2 sources is absolutely not well introduced and explained. For example, P5 L3-6 you should explain that these three sources of CO2 (atmospheric, limestone, organic C) have different isotopic composition. And you should give some order of magnitude maybe. The system of equations (9) should be carefully explained, in particular all the variables must be defined (what does Fm mean?).

RE: Modified accordingly. We reword this section to make it clearer to the reader and all variables are now properly defined (see P8 L15-29).

**(what does Fm mean?)**

RE: Fm is a defined radiocarbon metric relating to the 14C-to-12C ratio measured in the sample normalized to that of a standard. The metric was already defined in the original manuscript and appropriate references cited (P7 L21-24).

**Table 3 must include the isotopic composition of air of the site.**

RE: The information was already available in Table 2 (sample DRA17-ATM-2703) although we acknowledge that it was not clearly stated. We moved this piece of information to Table 3 to make it clearer to the reader.

**Results and discussions must be presented in distinct sections to clearly separate facts from their interpretations (and fit to the standard of BG).**

RE: We feel that a section "Results and discussion" is fitting for this manuscript – results are presented in Tables 2, 3 and 4 and in Figure 3. Other technical Note papers at Biogeosciences have the same format (e.g., Yoon et al., 2016; Call et al., 2017). Instead we added subsections (see sections 3.2.1, 3.2.2 and 3.2.3) and modified the title of section 3.3 to make it more informative (now "3.3. First order comparison of the magnitude of our  $CO_2$  fluxes with other methods estimating  $CO_2$  fluxes"). An introductive paragraph is added to this section to highlight some limitations of our study and justify our comparison with other indirect methods.

**To my opinion, the estimation of released CO2 by short-term measurement should be compared to the amount of CO2 trapped by zeolite. If this is not comparable, I expect detailed explanations of reasons.**

RE: We assume that the Reviewer refers to any difference between the active trapping  $CO_2$  flux measurements, and the mass of  $CO_2$  on the passive trap (accumulated over ~3 months). We are grateful to Reviewer 1 for an opportunity to expand on this interesting question. Before answering, let's relate the passive trapping to short-term flux measurements.

Passive sampling is a practical application of the first Fick's law (Bertoni et al., 2004). In our case it is related to the diffusion (D) of CO2 molecules in air caused by the gradient of CO2 partial pressure between that of the chamber ( $pCO_{2,Ch}$ ) and that of the zeolite trap ( $pCO_{2,zeolite}$ ). This diffusion is defined for a period of time ( $\Delta$ t) and is limited to the internal side of the tube linking the chamber to the zeolite trap, i.e. the diffusion path characterized by the tube length (L) and section area (a). It results in the trapping of a certain mass of carbon (mc) in the zeolite trap. In this case, first Fick's law may be written as follows:

$$pCO_{2,Ch} - pCO_{2,zeolite} = \frac{m_C}{\Delta t} \frac{L}{aD} \frac{RT}{PM_C} 10^6$$
(R1-1)

where R is the gas constant, T is temperature, P is pressure and  $M_c$  is the molar mass of carbon. Factor  $10^6 \times RT/PM_c$  converts grams of carbon to cm3 of CO2, and pCO2 is here in ppm (cm3/m3). Note that the pCO2,zeolite in the zeolite trap is equal to 0 ppm, since the zeolite is the CO2 absorber. The equation thus reduces to:

$$pCO_{2,Ch} = \frac{m_C}{\Delta t} \frac{L}{aD} \frac{RT}{PM_C} 10^6$$
(R1-2)

Equation (R1-2) allows us to reconstruct the average partial pressure of  $CO_2$  in the chamber  $pCO_{2,Ch}$  during the sampling duration ( $\Delta t$ ). Eq. R1-2 also indicates that the passive trapping is only directly linked to the partial pressure in the chamber over  $\Delta t$ . In other words, passive sampling is not related in a simple way to the flux of  $CO_2$  entering the chamber.

The above text should partially answer the Reviewer's comment. However, we can try to go further.

Let's assume that the evolution of the  $pCO_2$  in the chamber can be described (as we do in our manuscript for short-term flux measurements; see Eq. 7-8) following an exponential law (Pirk et al., 2016), we can express the  $pCO_{2,ch}$  in the chamber based on other parameters:

$$pCO_{2,Ch} = \frac{1}{V_{Ch}} \left[ \frac{q}{\lambda} \left( 1 - \exp(-\lambda \Delta t) \right) + m_0 \right] \frac{RT}{PM_C} 10^6$$
(R1-3)

where Vch is the volume of the chamber, q is the initial rate of carbon accumulation in the chamber, m0 is the initial mass of carbon in the chamber (a value that corresponds to 400ppm of CO2 in the volume of the chamber).  $\lambda$ , per unit of time, is the parameter that describes the diffusive processes responsible for the non-linear accumulation of carbon in the chamber (e.g. Fig. 3 in the manuscript). Note that  $\Delta t$  is very large (~3 months and thus ~150,000 minutes), thus exp( $-\lambda\Delta t$ )~0, and Equation R1-3 simplifies to:

$$pCO_{2,Ch} = \frac{1}{V_{Ch}} \left[ \frac{q}{\lambda} + m_0 \right] \frac{RT}{PM_C} 10^6$$
(R1-4)

Note that Eq. R1-4 can be written only if we assume that the initial rate of carbon accumulating in the chamber (q) does not change over time. This is a very large assumption that we expect to be violated because q is unlikely to stay constant over time for various reasons including natural variability in  $CO_2$  production and also changes in the diffusive processes when  $pCO_2$  builds up in the chamber.

Equating Eq. R1-2 and Eq. R1-4 we obtain:

$$\frac{m_{C}}{\Delta t} \frac{L}{aD} \frac{RT}{PM_{C}} 10^{6} = \frac{1}{V_{Ch}} \left[ \frac{q}{\lambda} + m_{0} \right] \frac{RT}{PM_{C}} 10^{6}$$
(R1-5)

Hence we can derive the rate at which carbon accumulates into the chamber based on the passive trapping parameters and  $\lambda$ , which is measured in the field over short time periods (during the active trapping – see Eq. 7-8 in the main text):

$$q = \lambda \left(\frac{m_{\rm C}}{\Delta t} \frac{L}{aD} V_{\rm Ch} - m_0\right) \tag{R1-6}$$

The flux can be inferred from the later equation using the internal surface area of the chamber ( $S_{Ch}$ ; same as S in the main text). If q was in gC/min, then the flux of carbon Q in gC/m2/year is:

$$Q = \lambda \left(\frac{m_c}{\Delta t} \frac{L}{aD} V_{Ch} - m_0\right) 525600 / S_{Ch}$$
(R1-7)

We can determine most of the parameters of equation (6) independently from the flux (Q or q), except for  $\lambda$ . For instance mc, m0, Vch, Sch,  $\Delta$ t, a and L can be measured and D (diffusion of CO2 in air) can be inferred/estimated from the literature. However,  $\lambda$  is determined using the short term flux measurements, along with the flux (i.e., Soulet et al. response to Reviewer 1, for "Technical Note: *in situ* measurement of flux and isotopic composition of CO2 released during oxidative weathering of sedimentary rocks"

Q or q). Thus estimating the flux of  $CO_2$  based on the rate of carbon trapped in the passive trap (mc/ $\Delta$ t) is not independent from the short term  $CO_2$  flux measurements. Thus comparing the fluxes obtained from the mass of carbon mc recovered using the passive trap and using Eq. R1-7 and the direct measurement, is somewhat circular because they are not determined independently from each other.

For longer monitoring of field work sites, the mass of carbon trapped is still informative but only qualitatively since  $m_c$  or better else the rate carbon trapping per unit of time ( $m_c/\Delta t$ ) are proportional to the flux of carbon Q to the chamber and the "leakiness" parameter  $\lambda$ . This is illustrated easily by writing equation Eq. R1-7 differently:

$$m_{C}/_{\Delta t} \propto Q/_{\lambda}$$
 (R1-8)

Interpretations of changes in  $m_c/\Delta t$  are thus qualitative at this stage, and so beyond the scope of the present Technical Note.

In the revised version we added some discussion clarifying that, based on our current knowledge and measurements, passive traps can be used to provide qualitative constraint on mass fluxes over time (see revised manuscript P10 L1-8).

We propose that future work investigates whether the parameter  $\lambda$  can be characterised for a chamber independently from the active CO2 flux measurements. If it can, the passive trap method can be used not only qualitatively (e.g. to look for changes in the mass of CO2 collected on passive traps through time), but quantitatively (i.e. the monthly time-integrated CO2 flux).

Title of section 3.3 is not clear and does not reflect the content. It seems that the objective of this section is to make a first comparison of estimation made by this study with published results from other sites.

RE: Indeed the objective of this section is to provide a first-order comparison between our results and other indirect river-catchment scale estimates of CO2 fluxes from around the world. We changed the title of this section to:

"3.3 First order comparison of the magnitude of our CO2 fluxes with other methods estimating CO2 fluxes"

If I understand well, your estimated amounts are far above the ones present in the literature. You should give some interpretations of these differences including the fact that your method has some bias that could lead to overestimations. First, the drilling can generate hyperactive surface by providing dust (small particles with high surface areas). Second, fresh surface is rich in organic C and limestone (because not previously exposed to O2). The CO2 may diffuse from pores of surrounding rock to chamber signifying that the surface of rock contributing to these estimations is larger than the sole surface of chamber.

RE: We partly replied to this comment in the Reviewer's detailed point P11L16. Regarding the additional specific points raised here by the reviewer:

- 1) Hyper-reactive surface by providing dust while drilling. The weathered marls in which we drilled are compact at depths greater than ~10cm (Mathys and Klotz, 2008; Osstwoud Wijdenes and Ergenzinger, 1998). But they are not extremely hard rock, and the chamber was drilled in about 1 or 2 minutes producing a coarse powder (that we actually needed to further grind to fine powder in the lab for our geochemical analyses). So we don't think that this powder was extremely reactive. Furthermore, before sealing the chambers the powder left inside the chamber was blown away using a compressed-air gun to minimize this phenomenon. We added a sentence in the revised manuscript stating we removed the powder before sealing (see revised manuscript P4 L19-20).
- 2) The rock surrounding the surface of the chamber contributes to the CO2 flux. We agree that the rock surrounding the surface of the chamber contributes to the CO2 flux, just because the weathering process naturally occurs at some depth within the rock face (probably in the regolith where gas can

penetrate through cracks). We realize that we might not have been clear enough about this point. What we call "rock" in the original manuscript should actually be referred to as the regolith which extends to up to ~20cm depth (Mathys and Klotz, 2008; Osstwoud Wijdenes and Ergenzinger, 1998; Maquaire et a., 2002). Thus when we drilled the chamber, we created a headspace, into an "ongoing-weathering" rock, in which CO2 can accumulate. This makes us able to measure a CO2 flux when we lower the pCO2 to that of the atmosphere. This net flux is the one we want to measure.

We added this information in the Study Area section (revised manuscript P4 L2-9) as well as in the Results and discussion section 3.1 (revised manuscript P9 L23-27).

**Detailed comments:**

**P3L11: why did you set up these two methods of trapping? The idea must be introduced before.**

RE: We dropped the mention of the two methods of trapping from the introduction. Furthermore the rationale behind using active and passive trapping was already explained in the method section of our original and revised manuscript.

**P4L4. The rock-drill was used to dig a hole or a cylinder. This is only a part of the chamber.**

RE: The rock-drill was used to drill directly into the rock/regolith a cylindrical chamber – 40cm-deep with an inner diameter of 2.9 cm. This is indeed only the headspace of the chamber. The rest of the chamber including how it is closed and sealed is described in the following lines (P4 L6-13 of the original manuscript)

**P4L24-27 The drilling makes powder that can stay on the surface. I guess that dust is highly reactive compared to rock that, has already been exposed to weather and oxidation since many years. This should be stated and discussed somewhere, maybe in the discussion section.**

RE: A relatively coarse powder is produced during drilling (with some coarse flakes). However, the rotating flute of the drill bit carries away most of the powder out of the hole/chamber. This allowed us to sample the powder to measure its organic and inorganic carbon contents and isotopes. Importantly, before sealing the chamber, the rock powder left in the chamber was blown away from the inside using a compressed air gun. This should minimize the impact of dust on the measured bulk CO2 flux. A sentence stating this point was added in the revised version of our manuscript (P4 L19-20).

**P6 L11-14 This text has no meaning for me, could you try to better explain? Concerning this section on the estimation of CO2 release, how did you manage the fact that released CO2 can accumulate in water present in the rock under the form of carbonates?**

RE: Indeed, a part of the  $CO_2$  could be dissolved in the interstitial water under the form  $H_2CO_3$  to form carbonate anions, of which a part could be released to the Laval stream water (the stream that drains the catchment where we installed the chambers). However, we are actually interested in measuring the net flux of  $CO_2$  to the atmosphere. Plumbing the whole system, i.e., quantifying the portion of  $CO_2$  that is emitted to the atmosphere as well as that that is released to the Laval stream as the form of carbonates anions after redissolution or as a result of Eq. 3 is out of the scope of this study.

Equations 9. You must say that this system of three equations can calculate three unknowns: atmopsheric CO2, limestone originating CO2, organic C originating CO2. Define all variables. Results and discussions must be separated. That will clarify your results and explanations.

RE: These comments are related to comments already addressed above.

P8L18-20. Not necessary, the dissolution of CO2 in water and formation of carbonates could lead to a non linear response.

Soulet et al. response to Reviewer 1, for "Technical Note: *in situ* measurement of flux and isotopic composition of CO2 released during oxidative weathering of sedimentary rocks"

RE: We do agree in principle. However rock water content seems to be very low as it does not look wet and we don't see any water dripping at all. So we doubt that the process described here by the reviewer can impact significantly the changes in pCO2 we observe repeatedly during our sequences of CO2 monitoring. Instead we keep thinking that a host of diffusive processes (Pirk et al., 2016; Kutzbach et al., 2007) are most likely to explain these non-linear changes.

P9L1-2 This decrease could result from an exhaustion of CO2 of pores surrounding the chambers (at the beginning of measurement these pores contribute much to the accumulation of CO2 in the chamber and they become empty with time).

RE: We do agree. This is what we explained at P9 L6-9 in the original manuscript.

P9L17 I disagree. The amount of atmospheric CO2 is given by your system of equations (9) RE: The manuscript text was correct as written.

Third equation of the system of equations (9) is:

 $f_{Atm} \cdot Fm_{Atm} + f_{RockOC} \cdot Fm_{RockOC} + f_{Carb} \cdot Fm_{Carb} = Fm_{Chamber}$ (R1-8)

Since rock-derived organic carbon and the carbonates are devoid of radiocarbon (because they are very old –  $^{160,000,000}$  years old, whereas  $^{14}$ C is not measurable anymore after  $^{50,000}$  years), hence their radiocarbon activity is 0. This implies that:

| $Fm_{RockOC} = Fm_{Carb} = 0$          | (R1-9) |
|----------------------------------------|--------|
| Combining (R1-8) and (R1-9), it comes: |        |

| $f_{Atm} = Fm_{Chamber} / Fm_{Atm} $ (R1- |
|-------------------------------------------|
|-------------------------------------------|

Thus the relative amount of atmospheric CO2 (fAtm) is calculated as written at P9 L17 in our original manuscript.

**P9L20 Cite Table 4**

RE: Sentence at P9L20 refers to stable carbon isotopes reported in Table 2. So we guess that the reviewer meant Table 2 instead of Table 4. Accordingly we now cite Table 2 in the revised manuscript (P10 L19).

P10 L10-11 You should better explain why do you make a correction for atmospheric CO2 although this contribution was already considered in your system of equations? After reading Table 4, I understood but you should better explain in the text.

RE: Modified accordingly, we added the specific information (P11 L31 to P12 L3 in the revised version)

**P10 L21-22 Change titles, they are not helpful for the understanding. RE: We already addressed this comment (see above)**

**P11 L3 You should add text to explain that your method must be compared with other (direct or indirect) methods on the same site.**

RE: This is the aim of section 3.3 where we compare our method of direct CO2 flux measurements to other estimations of CO2 fluxes. In the revised version, we hope the title of section 3.3 is now clear enough.

**P11 L5-9. The logical link between these sentences is not obvious.**

RE: Modified accordingly, see P13 L18 in the revised manuscript: "*This statement is supported by the average anion* [...]"

**P11L16 the numbers "19 to 37 gC m-2 yr-1" must be compared to "206 gC m-2 yr-1" of your study? The difference is enormous and deserves some explanations.**

RE: We agree the difference is large. We already provided some explanations at P11L18-20 of the original manuscript.

We could have added that seasonality might be another explanation, as CO2 flux during winter months flux could be very different than that during summer months as a results of changes in temperature and water content with impacts on the kinetics of rock weathering. We actually expect that the direct CO2 flux measurements would change over the course of the year. The estimates using dissolved calcium in river average several months. So one cannot expect that our direct flux CO2 measurements (for 1 discrete location on a given day) perfectly match the CO2 flux estimate using Laval stream chemistry (0.8 km2 averaging several months). Taken individually these CO2 fluxes are not comparable. What we felt important was to show that in erosive environments both fluxes are high compared to other geochemical carbon transfers (e.g. silicate weathering CO2 consumption).

We do not expand much on these explanations as our manuscript is a Technical Note. So this kind of discussion goes beyond the aim we set to our original manuscript. As Reviewer #2 notes, we explain a method that can now be installed more widely to explore these questions.

**Figure 1: I do not see what pictures B and C bring to the story.**

RE: We felt important to show a larger view of the field, as well as that the chamber are not drilled in soil horizons. In the case other reviewers would suggest that these pictures are superfluous, we will be happy to take them off of the paper.

**Have you checked that all the materials you use, especially the products used to seal and make tight (e.g. expansive foam etc), does not emit CO2?**

RE: We don't use expansive foam. Instead, we used outdoor silicon sealant (see P4L11 in the original manuscript). First we checked that the silicon sealant was not containing any curing agent like acetic acid, which may chemically alter the substrate around the hole. Second the flux measurements were performed when the sealant was fully dry (see P4L14). Flux measurements presented here were performed on March 2017, i.e., 3 months after we installed chamber H6 in December 2016. We are very confident that the sealant we use has no impact on our direct CO2 flux measurements.

Soulet et al. response to Reviewer 2, for "Technical Note: *in situ* measurement of flux and isotopic composition of  $CO_2$  released during oxidative weathering of sedimentary rocks".

**We are grateful to Reviewer 2 for theses constructive comments and concerns. Below we address the comments raised and provide corresponding amendment done to the original manuscript.**

In their Technical Note entitled, *"in-situ* measurement of flux and isotopic composition of CO2 released during oxidative weathering of sedimentary rocks", Soulet et al. report the results of a "proof-of-concept" study aimed at determining the release rate of carbon dioxide from outcrops of weathered shales and partitioning this carbon dioxide between inorganic and organic sources using C isotopes. The authors clearly describe the design, implementation, and data analysis for their rock weathering chambers in such a way that I am confident that I, or any other researcher, could implement this technique elsewhere. While I ultimately think that this paper should be published in *Biogeosciences*, I have a few comments that I would like to see the authors address (described below).

Firstly, I am confused by the distinction between a "direct" and "indirect" measure of a chemical weathering reaction. On Page 1 line 12, the authors imply that tracking reaction products (e.g., dissolved sulfate in rivers) is an indirect method. However, as carbon dioxide is also a reaction product, I do not see how their method is any more direct than measuring sulfate concentrations. Moreover, the relationship between the amount of product consumed (carbonate or organic carbon) and the amount of carbon dioxide release can be strongly modulated by the buffering capacity of natural waters. As a result, tracking carbon dioxide release may lead to a different assessment of the extent of reaction relative to a product that doesn't partition into both the fluid and gas phase (e.g., sulfate ion). That being said, I do agree that their method provides a different perspective on weathering reactions than measuring the dissolved or solid phase chemistry of rivers. In particular, I think the Soulet et al. method averages over very different temporal and spatial scales (see below) that make it a nice complement to river based approaches. Perhaps some more clarity as to what the authors mean by direct versus indirect would be helpful.

RE: The words "direct" and "indirect" were used to refer to the way the flux of CO2 emitted to the atmosphere during oxidative weathering of rocks has been measured/estimated in the literature. In other words, whether CO2 was being tracked directly, or by another product of the reaction (e.g. Re, or SO4) which is what we meant by 'indirectly'. However, we acknowledge that in two occurrences it was not clearly stated and agree with some of the reviewer's comments above. We have thus modified the manuscript accordingly (see P1 L11 and P2 L19 in the revised manuscript).

The different stoichiometries for carbonate weathering by sulfuric acid (CWSA) presented as equations 2 and 3 have appeared elsewhere in the literature. However, I am not convinced that, in the context of this paper, there is a real distinction that can be made. The dissolution of one mole of calcium carbonate releases one mole of calcium ion ( $Ca^{2+}$ ) and one mole of dissolved inorganic carbon (DIC;  $2H^+ + CaCO_3 \rightarrow Ca^{2+} + H_2CO_3$ ), which is equivalent to 2 units of alkalinity per unit of DIC. The generation of sulfuric acid from pyrite oxidation can titrate these 2 units of alkalinity leading to a net reaction for CWSA that results in 0 units of alkalinity generation per unit of DIC generation ( $CaCO_3 + H_2SO_4 \rightarrow Ca^{2+} + SO_4^{2-} + H_2CO_3$ ; equivalent to Equation 2 of Soulet et al.). Equation 3 of Soulet et al. predicts 1 unit of alkalinity generation per unit of DIC generation. In this way, it can be viewed as a 50/50 mixture of carbonate weathering by carbonic and sulfuric acids instead of a distinct pathway for CWSA.

RE: We agree with Reviewer 2. However, we make the distinction between these two pathways in our manuscript in order to link our work to a wider problem that includes the impact of oxidative weathering of rocks (including CWSA) on the CO2 concentration of the atmosphere over different timescales. We feel it can be better understood for a wider community using these two (and too) simple equations: Equation 2 implying the "immediate" release of CO2 to the atmosphere, and Equation 3 implying the release of CO2 over the timescale of 10,000 to 1,000,000 years. Depending on the fluxes involved, these pathways could thus impact climate over different timescales.

Furthermore, the idea that Equation 2 reflects an "immediate" release of carbon dioxide to the atmosphere misses the fact that the aqueous chemistry of weathering fluids will strongly modulate this flux. If there is sufficient generation of alkalinity from silicate weathering, the carbon dioxide produced from CWSA will partition more into the dissolved phase despite generally following the stoichiometry of Equation 2. Similarly, springs developed in carbonate terrains that lack abundant pyrite still degass carbon dioxide into the atmosphere despite the fact that the reaction for carbonate weathering by carbonic acid is often written as generating

bicarbonate ion. In other words, without more constraints on the fluid composition, it difficult to directly relate the extent of an individual weathering reaction to changes in carbon dioxide concentrations (e.g., see Soetart et al. 2007 Maine Chemistry).

RE: In the context of our study, "immediately" has to be compared to the timescales of 10,000 to 1,000,000 years. We acknowledge that in details aqueous chemistry of weathering fluids may modulate the  $CO_2$  flux, but from a geological point of view (104 to 106 year), this flux of  $CO_2$  is an "immediate" response to oxidative weathering of rocks.

In general, this study lacks replication. While I do not think that this is a critical issue, it'd be worth acknowledging some of the limitations and/or adding more analysis where possible. For example, two chambers are shown in Figure 1C. Is there not two chamber's worth of data to show?

RE: We do acknowledge that our methodology lacks replication, that's why we attempt comparing our results to other methods on other catchments despite issues of scales (see below). However, we are working in natural settings and we expect changes in the CO2 flux and isotopes in response to seasonal physical-meteorological changes in the catchment area. So we do not expect to find the exact same results for a single chamber over time, and for different chambers at the same time. Please also see our reply to Reviewer #1.

Based on both reviewers' comments we added a section regarding the limitation of our methodology (section 3.3 in the revised manuscript P12 L15-27).

**Similarly, I am not sure if I found at what depth below the land surface the chamber was placed. Presumably this depth will have a large effect on the results. What depth was selected and why?**

RE: We drilled the chambers on bare rock outcrops, and in places where we could not see roots (see Fig. 4). These outcrops make up 68% of the surface area of the catchment (Mathys et al., 2003; Cras et al., 2007) and are key parts of the landscape contributing to weathering, solute production (Cras et al., 2007) and sediment production (Mathys et al., 2003; Graz et al., 2012). These pieces of information were added in section 2.1 of the revised manuscript (P3 L25-32 and P4 L1-9).

Then, chambers are in the unsaturated zone, and the depth at which they were drilled depended on the accessibility in the field. Chamber H6 was drilled at ~2 meters above the Laval stream (P8 L3 in the revised manuscript).

Page 2 Line 24 - There are many other papers that have used S (and O) isotope ratios to partition the sulfate budget including some that precede the Calmels et al. 2007 paper. For example:

- \* Cameron, Eion M., et al. "Isotopic and elemental hydrogeochemistry of a major riversystem: Fraser River, British Columbia, Canada." Chemical geology 122.1-4 (1995): 149-169.
- \* Spence, Jody, and Kevin Telmer. "The role of sulfur in chemical weathering and atmospheric CO2 fluxes: evidence from major ions,  $\delta$ 13CDIC, and  $\delta$ 34SSO4 in rivers of the Canadian Cordillera." Geochimica et Cosmochimica Acta 69.23 (2005): 54415458.
- \* Das, Anirban, Chuan-Hsiung Chung, and Chen-Feng You. "Disproportionately high rates of sulfide oxidation from mountainous river basins of Taiwan orogeny: Sulfur isotope evidence." Geophysical Research Letters 39.12 (2012).

\* Turchyn, Alexandra V., et al. "Isotope evidence for secondary sulfide precipitation along the Marsyandi River, Nepal, Himalayas." Earth and Planetary Science Letters 374 (2013): 36-46.

\* Hindshaw, Ruth S., et al. "Influence of glaciation on mechanisms of mineral weather-ing in two high Arctic catchments." Chemical Geology 420 (2016): 37-50.

\* Torres, Mark A., et al. "The acid and alkalinity budgets of weathering in the Andes–Amazon system: Insights into the erosional control of global biogeochemical cycles." Earth and Planetary Science Letters 450 (2016): 381-391.

RE: Thanks. We added Spence and Telmer (2005) and Hindshaw et al. (2016) in the revised version of our manuscript.

Page 6 Line 15 - I'd appreciate a few sentences that explain directly how  $V_{ch}$  and S were determined. I assume that the dimensions of the drill hole and the assumption that it was shaped as a perfect cylinder were used. However, this ignores the fact the chamber walls are rough and not perfectly impervious. As a result, you are likely to get carbon dioxide from pores and cracks that intersect the chamber walls as alluded to on Page 9 Line 6. I'd appreciate some additional discussion on how this effects area-normalized estimates of carbon dioxide production rates.

RE:  $V_{Ch}$  and S were indeed determined assuming that the drilled hole is a perfect cylinder. We do agree that  $CO_2$  from a certain thickness around the drilled hole contributes overwhelmingly (compared to the  $CO_2$  flux produced at the rock-chamber interface) to the flux we measure.

However, it has to be noticed that we want to provide the community with a flux of  $CO_2$  emitted from the rock natural surface to the atmosphere. This flux includes the  $CO_2$  produced at the rock-atmosphere interface and the  $CO_2$  produced over a certain thickness from the weathered rock. In Draix, the thickness of the regolith is up to 10 to 20cm thick (Oostwoud Wijdenes & Ergenzinger, 1998; Mathys and Klotz, 2008). This means that the  $CO_2$  flux from the rock to the atmosphere is produced over a thickness of 10 to 20cm. Thus, when we drill a 40cm-long hole, rather than creating a new weathering surface at the rock-chamber interface, we instead create a headspace that makes us able to measure a realistic flux of  $CO_2$  from the rock to the atmosphere when we lower the p $CO_2$  to ~400ppm (atmospheric p $CO_2$ ).

We do agree that it was not clearly stated in the paper. We added some lines to state this point in the section 3.1 P9 L23-27 in the revised manuscript.

Furthermore, are their constraints from porosity, permeability, grain-size, and/or fracture density measurements that can inform the "effective" volume that the chamber samples? Or, could the mass of CO2 removed during the first few flushes inform this volume? Being able to determine an "effective" volume (as controlled by porosity, permeability, fracture density, grain size, etc.) would help others trying to replicate the methodology in determining if a site would be appropriate based on rock properties.

RE: There are some estimates of the rock properties for the Laval Catchment (Mathys et al., 2003; Oostwoud Wijdenes & Ergenzinger, 1998; Traveletti et al., 2002). These suggest i) the upper ~3 cm are loose material composed of mm-to-cm fragments of marls, ii) from ~3 to ~10cm is the regolith of marl more or less fragmented, iii) from~10 to 20 cm is the compact lower regolith keeping the marl structure but not its cohesion, and iv) the bedrock (unweathered marl). The porosity has been determined to be 0.17-0.23 (Traveletti et al., 2012). We have added these details to the study area part of the manuscript to help others seeking to replicate the methodology (Section 2.1 P4 L1-9).

However, what matters for the "effective" volume is the connected porosity and gas permeability which is, as the reviewer states, is probably linked closely to the fracture density. There are no measurements of this parameter at the field site and so we cannot use the mass of CO2 removed during the first flushes to inform us of this "effective" volume.

The purpose of our Technical Note is to show that one can measure reliable  $CO_2$  flux to the atmosphere using a cylindrical chamber and trap enough  $CO_2$  in the field to partition its source through its isotope composition (notably using 14C which requires larger volumes of  $CO_2$  to be collected). The controls on this flux (of which rock properties and connected porosity are likely to be one) cannot be assessed without more measurements at a range of chambers, and at a range of field sites.

Page 8 Line 27 - How realistic is it that the chamber has such a high  $pO_2$ ? My understanding of evidence from the oxygen isotopic composition of sulfate (e.g., Calmels et al. 2007), pyrite reactions fronts (Brantley et al. 2013 ESPL), and gas chemistry in wells (Kim et al. 2014 GCA 2017 GCA) is that oxidative weathering takes place under relatively low  $pO_2$  conditions for many systems. Does this mean that your method provides a maximum estimate of reaction rates?

RE: Weathering occurs not only at the atmosphere-rock interface but over at least a certain thickness into the rock (Petsch et al., 2000; Bolton et al., 2006). Nevertheless, at the atmosphere-rock interface  $pO_2$  is that of the atmosphere. From the chamber point of view, a  $pO_2$  of that of the atmosphere replicates what occurs when the rock is exposed to the atmosphere while  $pO_2$  probably decreases in depth in the rock.

Soulet et al. response to Reviewer 2, for "Technical Note: *in situ* measurement of flux and isotopic composition of CO2 released during oxidative weathering of sedimentary rocks".

It has to be noticed that our field site is not comparable to those described in the citations provided by Reviewer 2. In the suggested studies, erosion is much lower, leading to a very thick weathering front of 20 m or more (e.g., Brantley et al., 2013). In Draix – our field study – erosion removes 1cm of rock in average per year but it can be more. So the weathering front is probably far less thick, and  $pO_2$  higher.

Page 9 Line 9 - For the analysis of  $CO_2$  fluxes, it is stated that 3-4 flushes are necessary to get the "true" flux determination. What statistical criteria was this determination based on? Similarly, what is the basis for designating 6 minutes as the amount of time to fit the carbon dioxide accumulation curve (Page 6 Line 11)?. How do the calculated averages and standard deviations of  $CO_2$  flux vary with measurement / integration time?

RE: When the flux measured vs. number of repeats is examined (Figure R1), one observes a decrease in the flux that reaches steady values after 3 to 4 repeats, while statistically the last four are indistinguishable within  $2\sigma$ . We are deliberately vague in our manuscript as this feature can change depending on the chamber and flux. In practice, the number of repeats on which flux is averaged has to be adapted based on the results observed.

Figure R1: Evolution of the measured flux with the number of repeats (grey filled squares, error bars are 2σ). Dashed line is the averaged flux over the last 4 repeats (257±8 gC/m2/yr) and yellow bar represents the 2σ-domain of the averaged flux.

Regarding the window for the flux measurements. If we pick 1 to 8 minutes of fitting, the results all agree within 2 sigma (Figure R-2).

The fitting window has to be specified and in our case, 6 mins were chosen as a trade-off. In our manuscript, we present a series of active trapping for which we left the chamber replenish with CO2 for more than 6 mins. However, when on field, we are not necessarily trapping CO2. Instead we are sometimes only measuring fluxes. In these cases, for logistical reasons mainly related to the time we can spend in the field daily, we are monitoring over shorter periods of 7 minutes. Thus fitting over 6 mins was sensible. This parameter can be modified as soon as it is specified. For example, Pirk et al (2016) chose 3-minutes fitting windows. The starting point of the fitting window may also impact slightly the results, although providing similar results within 2 sigma, if it is set to pCO2 close to atmospheric values.

Soulet et al. response to Reviewer 2, for "Technical Note: *in situ* measurement of flux and isotopic composition of  $CO_2$  released during oxidative weathering of sedimentary rocks".

Figure R2: Evolution of the average flux (over the last 4 repeats) with changing fitting time windows from 1 to 8 minutes (grey filled circles, error bars are 2σ). All data agree within 2σ. Please, note that scale of y-axis is different from Figure R1.

Page 9 Line 20 - I am not convinced that the difference between the 2 carbon isotopic samples reflects process and not fractionation. The analysis of carbon dioxide fluxes explicitly assumes that there are leaks in the system, which may induce fractionation. Similarly, two different methods were used for these samples. Finally, if the balance between oxidative reactions can vary daily, then why is the entire difference in the isotopic composition of CO2 derived from the passive trapping method assumed to result from fractionation. In general, a better discussion of which isotopic signals are attributed to environmental process vs. sampling-induced fractionation and why would be helpful.

RE: Previous studies when developing the passive method quantified an isotopic fractionation (Garnett et al., 2009; Garnett and Hardie, 2009; Garnett and Hartley, 2010). In contrast, the pump/active method doesn't fractionate, as shown by e.g. Hardie et al 2005.

We agree that our discussion about fractionation was a bit short. We expanded this discussion and highlighted potential limits but also benefits of using the passive and active sampling methods.

Please note that we changed the  $3.5 \pm 0.45$  ‰ fractionation values (based on merging values provided in Garnett et al., 2009; Garnett and Hardie, 2009) by the now accepted value of  $4.2 \pm 0.3$  ‰ (Garnett and Hartley, 2010) based on a laboratory assessment. This value is indistinguishable from the value obtained in Garnett and Hardie (2009) of  $4.0 \pm 0.2$  ‰. The new applied value minimally changes the source partitioning results and does not changes our interpretations.

Page 10 Line 15 - **This is very interesting!** In other words, the release ratio of inorganic to organic carbon determined by carbon isotope ratios is different than the relative abundances of inorganic and organic carbon present in the underlying rock. Specifically, the isotopic method "sees" more organic carbon than would be expected if one "unit" of rock was congruently weathered. Does this make sense with what is known about carbon and sulfur reaction fronts in weathering profiles?

RE: Once again, in Draix the weathering profile is probably thin (several decimeters) compared to other weathering profiles published (several meters; e.g., Brantley et al., 2013). At this stage, we reiterate the explanation we provided in the original version of our manuscript (P10 L18-20). The dissolution of carbonate depending on the oxidation of sulphides, it is therefore likely that it only occurs locally where sulphides are

Soulet et al. response to Reviewer 2, for "Technical Note: *in situ* measurement of flux and isotopic composition of  $CO_2$  released during oxidative weathering of sedimentary rocks".

concentrated. In comparison the oxidation of organic carbon appears to occur homogeneously in the rock mass. We agree it is interesting and worthy of future study.

Page 10 Line 21 - While I understand the motivation behind including section 3.3, I think that the different area normalization schemes between the chambers and river-based measurements precludes direct comparison. The area normalization in river systems refers to the catchment area. However, weathering takes place at depth within porous media such that the true surface area of reactive material that rivers source solutes from is likely poorly approximated by the catchment surface area. In the chamber experiments, the area normalization refers to the surface area of the chamber walls, which likely more closely approximates the true "reactive" surface area (see above). At the very least, this discrepancy between area normalization schemes should be discussed before generating comparisons between the different datasets. Depending on how reactive surface area scales with catchment area, the fact that the chamber-based estimates are close in magnitude to the river-based estimates may actually mean that there is a large discrepancy in the rates that they predict.

RE: We somewhat agree and have added discussion to the revised version which relates to the referee's comment. This is a common issue when referring to element fluxes per unit of surface area. With more CO2 flux measurements, alongside solute-based weathering estimates, we will be in a better position to probe these differences in more detail.

**Technical note: *in situ* measurement of flux and isotopic composition of CO2 released during oxidative weathering of sedimentary rocks**

Guillaume Soulet1, Robert G. Hilton1, Mark H. Garnett2, Mathieu Dellinger1, Thomas Croissant1, Mateja Ogrič1 and Sébastien Klotz3

1Department of Geography, Durham University, South Road, Durham DH1 3LE, United Kingdom 2NERC Radiocarbon Facility, Rankine Avenue, East Kilbride, Glasgow G75 0QF, UK 3IRSTEA Grenoble, 2 rue de la papeterie, BP 76, 38402 Saint-Martin-d'Hères, Cedex, France *Correspondence to*: Guillaume Soulet (guillaume.s.soulet@durham.ac.uk)

Abstract. Oxidative weathering of sedimentary rocks can release carbon dioxide (CO2) to the atmosphere and is an important

- 10 natural CO2 emission. Two mechanisms operate the oxidation of sedimentary organic matter and the dissolution of carbonate minerals by sulphuric acid. It has proved difficult to directly measure the rates at which CO2 is emitted in response to of these weathering processes in the field, with previous work generally using indirect methods which track the dissolved products of these reactions in rivers. Here we design a chamber method to directly measure CO2 production during the oxidative weathering of shale bedrock, which can be applied in erosive environments where rocks are exposed frequently to the atmosphere. The
- 15 chamber is drilled directly into the rock face and has a high surface area to volume ratio which benefits measurement of CO2 fluxes. It is -and is a relatively low cost method toand provides a long-lived chamber (several months or more), oxygenated environment in contact with a surface area of potential reactant. To partition the measured CO2 fluxes and the source of CO2, we use zeolite molecular sieves to trap CO2 'actively' (over several hours) or 'passively' (over a period of months). The approaches produce comparable results, with the trapped CO2 having a radiocarbon activity (Fraction modern, Fm) fraction
- 20 modern ranging from  $\underline{Fm} = 0.05$  to  $\underline{Fm} = 0.06$  and demonstrating relatively little contamination from local atmospheric CO2 (fraction modern of  $\underline{Fm} = 1.01$ ). We use stable carbon isotopes of the trapped CO2 to partition between an organic and inorganic carbon source. The measured fluxes of rock-derived organic matter oxidation (171±5 mgC.m-2.day4) and carbonate dissolution by sulphuric acid (534±17 mgC.m-2.day-1) from a single chamber were high when compared to the annual flux estimates derived from using dissolved river chemistry in rivers around the world. The high oxidative weathering fluxes are , but
- 25 consistent with the high erosion rate of the study region  $(of \sim 5 \text{ mm yr}^4)$ . We propose our in situ method has the potential to be more widely deployed to directly measure CO2 fluxes during the oxidative weathering of sedimentary rocks, allowing for the spatial and temporal variability in these fluxes to be determined.

**1** Introduction**

The stock of carbon contained within sedimentary rocks is vast, with  $\sim 1.25 \times 10^7$  PgC contained within organic matter and  $\sim 6.53 \times 10^7$  PgC as carbonate minerals (Sundquist and Visser, 2005). If these rocks are exposed to Earth's oxygenated surface,

**Commented [SGS1]:** We now provide the CO2 flux measurements in units of mgC/m2/day at the first instance. This is intended to clarify to the reader that our fluxes represent those over a short period of time, allowing us to be more cautious when we compare these first CO2 emission measurements to published work on oxidative weathering fluxes.

for instance during rock uplift, erosion and exhumation, oxidative weathering can result in a release of carbon dioxide  $(CO_2)$  from the lithosphere to the atmosphere (Petsch et al., 2000). There are two main processes to consider: i) the oxidation of rockderived organic carbon (Berner and Canfield, 1989; Petsch, 2014), which can be expressed by the (geo)respiration of organic matter:

$$\quad CH_2 0 + 0_2 \to CO_2 + H_2 0 \tag{1}$$

and ii) the oxidation of sulphide minerals (e.g., pyrite) which produces sulphuric acid, which can chemically weather carbonate minerals and release  $CO_2$  (Calmels et al., 2007; Li et al., 2008; Torres et al., 2014) by the reaction:

$$CaCO_3 + H_2SO_4 \to CO_2 + H_2O + Ca^{2+} + SO_4^{2-}$$
(2)

or

 $10 \quad 2CaCO_3 + H_2SO_4 \to 2Ca^{2+} + 2HCO_3^- + SO_4^{2-} \to CaCO_3 + CO_2 + H_2O + Ca^{2+} + SO_4^{2-}$ (3)

In the case of Eq. (1) and Eq. (2),  $CO_2$  is released to the atmosphere at the site of chemical weathering"<del>immediately</del>". In the case of Eq. (3),  $CO_2$  is released to the atmosphere over a timescale equivalent to that of the precipitation of carbonate in the ocean (~104 to 106 years; Berner and Berner, 2012).

- The fluxes of carbon transferred to the atmosphere in response to both oxidative weathering processes are thought to be as 15 large as that released by volcanic degassing, but the absolute fluxes remain uncertain (Li et al., 2008; Petsch, 2014). As such, both processes act to govern the levels of atmospheric CO2 and O2, and hence Earth's climate over geological timescales (Berner and Canfield, 1989; Torres et al., 2014). The oxidation of rock-derived organic carbon may also contribute to modern biological cycles, especially rock substrate that is rich in organic carbon (Bardgett et al., 2007; Copard et al., 2007; Keller and Bacon, 1998; Petsch et al., 2001). Various approaches have been adopted to better quantify these major geological CO2 sources.
- 20 These include the use of geochemical proxies in rivers, which indirectly track the CO2 emissions released-from the oxidative weathering of sedimentary rocks at the catchment-scale. For instance, the trace element rhenium has been used to compare relative rates of rock-derived organic carbon oxidation (Jaffe et al., 2002) and estimate the corresponding fluxes of CO2 across river catchments (Dalai et al., 2002; Hilton et al., 2014; Horan et al., 2017). Another approach has been to measure the loss of radiocarbon-depleted organic matter in river sediments during their transfer across the floodplains of large river basins
- 25 (Bouchez et al., 2010; Galy et al., 2008). In the case of sulphuric acid-weathering of carbonate minerals, the dissolved sulphate flux can be informative if the source of SO42- has been assessed using sulphur and oxygen isotopes (Calmels et al., 2007; Hindshaw et al., 2016; Spence and Telmer, 2005) and/or using the dissolved inorganic carbon flux and its stable carbon isotope δ13C-composition (δ13C) (Galy and France-Lanord, 1999; Li et al., 2008; Spence and Telmer, 2005).

It should be possible to directly measure the flux of  $CO_2$  emanating from sedimentary rocks in response to oxidative weathering. Keller and Bacon (1998) explored such an approach in a 7 m deep soil on till, suggesting geo-respiration of Cretaceous age organic matter was an important source of  $CO_2$  at depth. However, this research has not to our knowledge been replicated, nor applied in erosive landscapes where sedimentary rocks are frequently exposed to weathering by erosion processes (Blair et al., 2003; Hilton et al., 2011). In these locations, oxidative weathering fluxes are thought to be very high (Calmels et al., 2007; Hilton et al., 2014; Petsch et al., 2000). One of the challenges of tracking  $CO_2$  directly is that flux

measurements must be combined with the isotopic composition (12C, 13C and 14C) of the CO2 (Keller and Bacon, 1998). Only with that information can the measured CO2 flux be partitioned into the component derived from the oxidation of rock-derived carbon and that derived from the dissolution of carbonate (in addition to inputs from the modern plant and soil biosphere, and atmospheric inputs).

The objective of this paper is to provide a detailed proof of concept study of methods we have designed which can: (1) make direct measurements of the flux of CO2 released during the oxidative weathering of sedimentary rocks; and (2) trap the CO2 produced during weathering in order to measure its isotope composition, and partition the source of the CO2 flux between rock-derived organic carbon and carbonate. Here we outline one approach to address these research gaps which adapts a the

15 chamber-based method to measure CO2 fluxes. We ,- and provide the first examples of its application of active and passive methods to trap CO2 and use its-the isotope composition to directly quantify the fluxesrate of CO2 from oxidative weathering reactions.

**2 Methods**

5

**2.1 Study area**

[revised manuscript text omitted]

**20 3.1 Flux Measurementsmeasurements**

Three months after the installation of the chamber H6, CO2 fluxes were measured alongside a series of zeolite-trapping events on 27/03/2017 (Figure 3). If the chamber was perfectly isolated from the atmosphere, then we might expect the rate of carbon accumulation  $\left(\frac{dm(t)}{dt}\right)$  to be constant, while it decreases with time. As expected, this indicates that the chamber is not perfectly sealed. This has some important implications. First, the leak rate depends on the pCO2 gradient between the chamber and the atmosphere. Since this gradient is positive in the chamber (Figure 2)  $(pCO_{2chamber} > pCO_{2atmosphere})$ , then CO2 likely diffuses from the chamber to the atmosphere. This has the advantage that it naturally minimizes any atmospheric CO2 contamination. Conversely, since the CO2 production is linked to the consumption of O2, then the O2 gradient is expected to be negative  $(pO_{2chamber} < pO_{2atmosphere})$ , and thus atmospheric O2 naturally diffuses inside the chamber. This means that the chamber can be closed for months and still contain gaseous O2. Our measurements of O2 using the EGM-5 O2 probe suggest

that the chamber had a similar  $pO_2$  as that contained in the ambient atmosphere of the catchment (the chamber value was 96 to 99% of the atmosphere  $pO_2$ ).

The fluxes of CO2 measured in this chamber on 27/03/2017 decreased from  $529 \pm 14 \text{ gC.m}^2 \cdot \text{yr}^4 - \underline{1384 \pm 42 \text{ mgC.m}^2 \cdot \text{day}^{-1}}$ to  $\underline{266 \pm 7 \text{ gC.m}^2 \cdot \text{yr}^4} \cdot \underline{684 \pm 21 \text{ mgC.m}^2 \cdot \text{day}^{-1}}$  with the number of times we extracted the CO2 from the chamber (Figure 3).

- 5 The flux becomes approximately constant after three CO2 extractions during zeolite trapping, with an average of 272±8 gC.m-
  2·yr+ 705 ± 15 mgC.m-2.day-1 (1sd, n=4) for the last 4 flux repeatsmeasurements that are iundistinguishable from each other within 2σ (Figure 3). This observation might be explained by the fact that over time (days to months), CO2 accumulates not only in the chamber, but also in the regolith/rock connected pores surrounding the chamber in the lower compact regolith (Maquaire et al., 2002)surrounding the chamber. Weathering reactions are likely to occur not only at the chamber-rock interface, but also into the rock mass over several centimetres as the weathered regolith extends to depths of up to 20 cm
- (Maquaire et al., 2002; Mathys and Klotz, 2008; Oostwoud Wijdenes and Ergenzinger, 1998). When CO2 is first extracted from the chamber, the CO2 stored in the surrounding pores quickly refills the chamber. It appears that after three extractions this CO2 is depleted, meaning that the more constant values correspond to the actual flux of CO2
- through the surface area of the chamber. We would therefore recommend that flux measurements are made on such a chamber
  following ~3 to 4 removals of CO2, or adapted to less or more removals based on the results obtained after a series of flux measurements. It remains to be seen the degree to which this feature is widespread, or chamber specific.
- It has to be noted that the mass of carbon (mC) recovered on the zeolite molecular sieve during the period of passive trapping (Δt) cannot be directly and simply used to inform the flux of carbon through the chamber. This is because the rate of carbon trapping (mC/Δt) follows the first Fick's law (Bertoni et al., 2004) and hence depends on